# LEARNING GOAL-ORIENTED LANGUAGE-GUIDED NAVIGATION WITH SELF-IMPROVING DEMONSTRATIONS AT SCALE

## ABSTRACT

Goal-oriented language-guided navigation requires robust exploration capabilities for agents to navigate to specified goals in unknown environments without step-by-step instructions. Existing methods tend to exclusively utilize shortest-path trajectories, lacking effective exploration priors for training navigation agents. To address the above challenges, we present SID, a goal-oriented language-guided navigation learning approach with Self-Improving Demonstrations. Specifically, SID learns an initial agent on the shortest-path data sampled from environments and then leverages this agent to generate novel exploration trajectories. The novel rollouts provide demonstrations with stronger exploration strategies to train a better agent, which in turn produces higher-quality agent demonstrations for the next round of training. We show that this iterative self-improving pipeline readily scales to new environments, and the resulting demonstrations can be transferred across a variety of language-guided navigation tasks, elevating the performance ceiling in diverse goal-oriented navigation tasks. Extensive experiments demonstrate that SID significantly boosts the exploration capabilities and generalization of navigation agents. The resulting agent achieves new state-of-the-art performance on goal-oriented language-guided navigation tasks, including REVERIE, SOON, notably achieving a 50.9% success rate on the unseen validation splits of SOON, surpassing the prior leading approaches by a margin of 13.9%.

## 1 INTRODUCTION

Developing embodied agents capable of following human instructions to navigate the real world has been a longstanding goal in artificial intelligence. Research in this area encompasses various tasks, including Vision-Language Navigation (VLN) with step-by-step instructions (Zhang & Kordjamshidi, 2024) (*e.g.*, R2R (Anderson et al., 2018b), RxR (Ku et al., 2020)) and goal-oriented VLN tasks (Chaplot et al., 2020) (*e.g.*, REVERIE (Qi et al., 2020) and SOON (Zhu et al., 2021)). While step-by-step instructions like "*Go down the hallway, turn left at the kitchen, and stop in front of the fridge*" give detailed path descriptions, they are impractical for real-world applications where humans prefer concise commands, using Goal-oriented instructions like "*bring me a spoon from the kitchen*".

Given that goal-oriented instructions primarily depict the target and its surroundings, agents need to explore the environment to find it. Early efforts either utilize Reinforcement Learning (RL) for exploration (Mirowski et al., 2016; Zhu et al., 2017; Mousavian et al., 2019; Gupta et al., 2017), or learn a navigation policy from human demonstrations (Ramrakhya et al., 2022b). The former requires sophisticated reward engineering, suffering from unstable training and massive compute demands (Wijmans et al., 2020; Krantz et al., 2021), while the latter leans heavily on costly human annotations that are hard to collect and transfer to language-guided navigation tasks. Recently, with improved models (Vaswani et al., 2017) and large-scale synthetic instructions (Chen et al., 2022b; Wang et al., 2023b; Li & Bansal, 2023b; Wang et al., 2024b; 2025b;a), a number of methods (Chen et al., 2022a; Wang et al., 2023a; 2024a) leverage pure Imitation Learning (IL) to develop exploration policies. Despite these progresses, these models are typically exclusively pretrained by greedily imitating the shortest-path samples, leading to agents devoid of effective exploration priors. Although online finetuning aids in error correction during decision making, this exploration-agnostic pretraining inherently limits generalization in unseen environments.

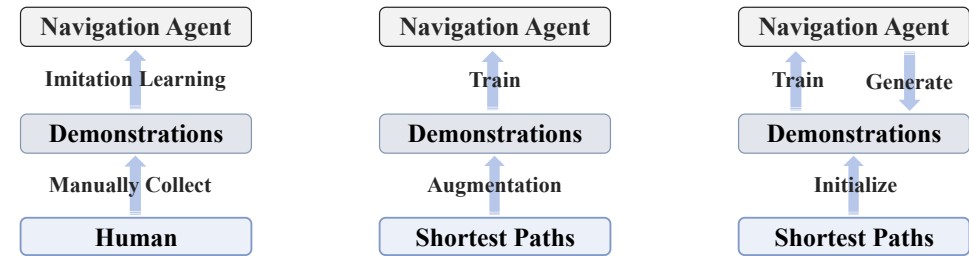

(a) IL on Human Demonstrations.     (b) Shortest-Path Augmentation.     (c) Self-Improving Demonstrations.

Figure 1: Comparison of three goal-oriented VLN training paradigms. (a): Learning exploration from imitating human demonstrations, which is costly and difficult to scale up. (b): Learning general navigation from large-scale instruction augmentation on shortest-paths, which lacks demonstrations on exploration. (c): Learning goal-oriented VLN with self-improving demonstrations. We initialize demonstrations with shortest paths and then use the trained agent to generate novel demonstrations, which are in turn leveraged to further improve the agent, forming a self-improving loop.

In this regard, we propose learning from **S**elf-**I**mproving **D**emonstrations(**SID**), a goal-oriented navigation training framework that leverages the agent's own successful exploration data as IL demonstrations, as shown in Figure 1. We start by applying SID to a classical goal-oriented navigation task, image-goal navigation—where data is easily acquired as each target image and starting viewpoint pair naturally forms a training sample—in base environments such as Matterport3D. Specifically, SID starts with training a base agent with shortest paths demonstrations in MP3D, then this agent infers new exploration trajectories (*e.g.*, explores different rooms and distinguishes fine-grained visual elements), and only successful ones are collected as self-demonstrations to train a more capable agent. The agent in the new iteration is pretrained on these self-demonstrations, followed by finetuning with both self-demonstrations and shortest-path data to effectively balance exploration and exploitation. The finetuned agent generates new self-demonstrations for training the next-iteration agent. This iterative cycle brings consistent performance gains, where improved agents generate superior exploration trajectories and, in turn enhance subsequent agent training.

When the agent's performance saturates within the initial environments after several iterations of self-demonstration learning, we extend SID by incorporating additional environments (*e.g.* HM3D (Ramakrishnan et al., 2021)) to expand the training corpus and continue the iterative SID training, resulting in a proficient navigation agent along with large-scale image-goal demonstration trajectories containing rich exploration priors. We then explore transferring SID trajectories to goal-oriented VLN tasks such as REVERIE and SOON. To address the scarcity of paired instructions for each target image, we propose a simple but effective augmentation method that leverages Vision-Language Models (VLMs) to generate detailed captions as instructions. The agent is first pretrained on the resulting caption-demonstration pairs and then finetuned on downstream tasks.

Extensive experiments show that the agent's self-demonstrations not only consistently outperform shortest-path demonstrations on both image-goal navigation and VLN tasks but also benefit both pretraining and finetuning. The agent improves steadily with higher-quality self-demonstrations generated by stronger agents at each iteration, and this loop can be continued by introducing more environments. With our final SID data, a simple navigation agent such as DUET (Chen et al., 2022a) achieves state-of-the-art performance on goal-oriented VLN tasks, including REVERIE and SOON. Notably, it reaches a 50.9% success rate on SOON, exceeding the previous approaches by a significant margin of 13.9%. Generally, our contributions can be listed as follows:

- We present **Self-Improving Demonstrations (SID)**, a novel iterative approach that enables a navigation agent to learn robust exploration strategies with demonstrations from its own successful trajectories, averting the dependence on costly human annotations for exploration data.

- We provide large-scale exploratory language-goal trajectories along with a capable navigation agent that can be used to generate additional exploration paths, which is the first to offer transferable large-scale demonstrations on exploration strategies, filling a critical gap in goal-oriented navigation.

- We demonstrate SID's scalability across diverse environments and transferability to goal-oriented language-guided navigation tasks through vision-language-aligned pretraining, achieving state-of-the-art results on challenging goal-oriented VLN tasks including SOON and REVERIE.

## 2 RELATED WORK

**Vision-and-Language Navigation.** Learning to autonomously navigate in new environments by comprehending and executing natural language instructions remains a central goal towards advanced agents in the field of Vision-and-Language Navigation. Currently, numerous scenarios have been proposed for VLN tasks, including navigation via detailed instructions of sequences of actions (Anderson et al., 2018b; Ku et al., 2020; Thomason et al., 2020), remote object navigation with coarse-grained high-level instructions (Zhu et al., 2021; Qi et al., 2020), and object-goal navigation in continuous environment (Kolve et al., 2017; Krantz et al., 2020; Xia et al., 2018; Hong et al., 2022). To approach this problem, previous methods concentrate primarily on LLM planning for VLN (Zhou et al., 2023; Lin et al., 2024; Zhou et al., 2024a), leveraging generic visual-linguistic representations and alignment (Chen et al., 2021; Guhur et al., 2021; Li et al., 2022; Li & Bansal, 2023b;a; Liu et al., 2024; Wang et al., 2024a), and enhancing navigation action planning mechanisms with map construction and historical memorization (Zhao et al., 2022; Wang et al., 2023a; Hong et al., 2021; Wang et al., 2020; Liu et al., 2023). Although there has been considerable progress in this field, the high cost of collecting human-annotated instruction trajectory data to achieve near-human performance is still a key obstacle to training generalizable capable agents for VLN. Numerous data augmentation methods have been proposed like training instruction-generator or extracting large-scale datasets from rendered environments in simulators (Hao et al., 2020; Zhang et al., 2024b; Ramakrishnan et al., 2021; Xia et al., 2018; Wang et al., 2023b; 2024b), but none of which satisfies the need for large-scale high-quality demonstrations on exploration strategies. To tackle this challenge, we propose self-improving demonstration method for goal-oriented VLN.

**Self-Improving Agents.** A self-improving agent is capable of autonomously enhancing its performance through iterative feedback to achieve continuous improvement without extensive human intervention (Lin, 1992; Singh et al., 2023). Significant progress has been made in various fields such as mathematical reasoning (Yuan et al., 2023; Zhao et al., 2025), software engineering (Hu et al., 2024) and robotic manipulation (Bousmalis et al., 2023). The quality of self-generated data can be improved through various techniques, including model's self-feedback (Wu et al., 2024; Yuan et al., 2024), reward ranked or rejection sampling finetuning (Dong et al., 2023; Zelikman et al., 2022; Xiong et al., 2025). However, in robotics tasks, complicated sampling strategies are impractical due to the intricate evaluation of alignment of instruction-trajectory pairs (Wang et al., 2024b), and the key obstacle is the quality of demonstrations rather than correctness (*e.g.*, both exploration trajectories and shortest paths are correct, but the former provides better demonstrations than the latter). In SID, the agent is capable of iteratively improving itself by leveraging its own generated trajectories, elevating the dependence on extra data generator and sophisticated negative sample processing.

**Learning with Demonstrations.** Learning from demonstrations to overcome inefficient exploration (Nair et al., 2018) is essential for sparse-reward tasks such as robotic navigation and manipulation. For instance, TCN (Sermanet et al., 2018) offers a approach to learning from video demonstrations for tasks requiring temporal understanding. RoboTube (Xiong et al., 2022) showcases the potential to learn household manipulation tasks from human videos demonstrations with simulated environments. More recently, RLDG (Xu et al., 2024) distills generalist policies from RL-generated trajectories to reduce human dependency and enhance adaptability. In goal-oriented navigation, Habitat-Web (Ramrakhya et al., 2022a) leverages human demonstrations to train exploration-aware agents for object navigation. Unlike these approaches, SID bootstraps from the agent's self-demonstrations iteratively, enabling strong exploration policy without external human effort.

## 3 SELF-IMPROVING DEMONSTRATIONS FOR GOAL-ORIENTED VLN

Goal-oriented navigation involves planning an agent (navigator) to reach the target only with its visual / textual description in unseen environments. Existing methods still rely on costly human demonstrations to devise feasible exploration strategies connecting to the target places. To address this data bottleneck, we propose **Self-Improving Demonstrations** to scale its training data in a semi-supervised manner. In the following, we detail this method and and its training corpus.

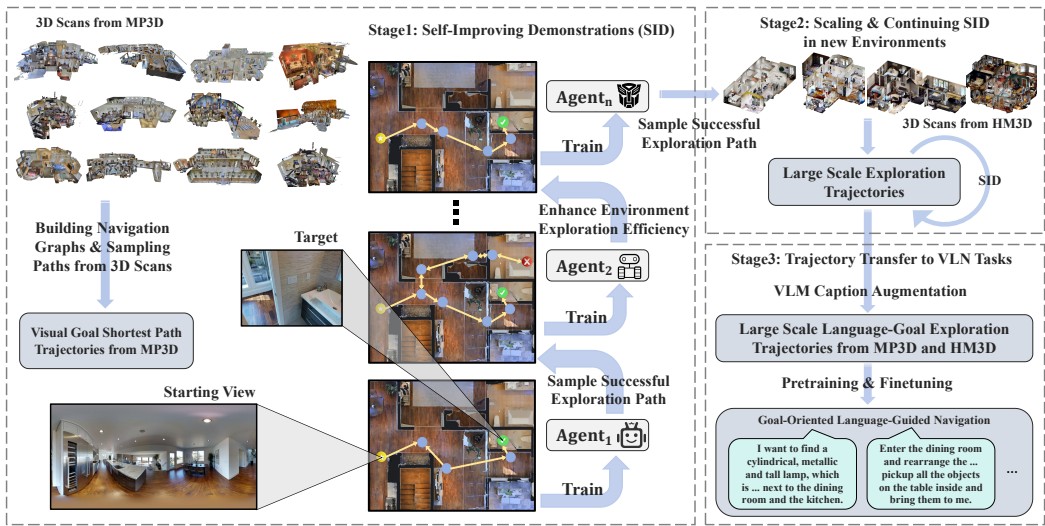

Figure 2: Our proposed Self-Improving Demonstrations paradigm for goal-oriented VLN. We learn an initial navigation agent using trajectories sampled from MP3D, generate new paths using this agent, and reserve the successful exploration ones. These trajectories give demonstrations on the exploration strategies, resulting in a more capable agent. This iterative semi-supervised learning can gradually improve navigation agent's performance ceiling and produce effective exploration trajectories at scale, which can be transferred with caption augmentation for goal-oriented VLN.

## 3.1 SELF-IMPROVING DEMONSTRATIONS ON IMAGE GOAL NAVIGATION

We give an approach that continuously improves the quality of training data through an iterative pipeline, as in Figure 2. This contains 1) building base training data $D_0$, 2) learning a navigator $\mathcal{N}_{\theta_0}$ with the data, and 3) generating new trajectories from the navigator. This repeats multiple rounds to optimize the agent's path-planning capabilities as well as goal-oriented navigation data quality. We empirically show it is convergent and consistently improves navigation performance in Section 4.

**Constructing Base Data.** We consider navigation in discrete environments. It is formulated as the path planning in graph theory (Anderson et al., 2018b) and each environment is treated as an undirected graph $\mathcal{G} = \{\mathcal{V}, \mathcal{E}\}$, where $\mathcal{V} = \{v_1, v_2, ...v_n\}$ denotes $n$ navigable viewpoints and $\mathcal{E}$ denotes connectivity edges. Each viewpoint has a panoramic observation $\mathcal{O} = ([\mathbf{I}_i; a_i])_{i=1,..,K}$ of its surroundings and a location coordinate representation $\mathbf{v} = (x, y, z)$. Note $\mathbf{I}_i$ is the $i$-th view (an image), $a_i$ is the relative angle to face it, and a panorama is composed of $K$ images.

We traverse all pairs of nodes in the graph and retain those with the corresponding paths similar to previous man-made datasets from MP3D. Specifically, we compute the shortest path between the viewpoints of each pair, and preserve those whose length are between 5 and 7 viewpoints. In this procedure, we construct a total of over 180,000 trajectories. We associate the panoramic observations to them and get visual-goal-trajectory pairs as raw training data $D_0 = \{(\mathbf{p}_i, \mathbf{g}_i,)\}_{i=1,...,n}$, where $\mathbf{p}_i = <\mathbf{v}_i^j>_{j=1,..,l_i}$ is a feasible path reaching the goal $\mathbf{g}_i$, which is the description (*e.g.* an image or a sentence capturing the destination) of the target location $\mathbf{p}_i[-1]$ (the end location of the path $\mathbf{p}_i$). $n$ and $l_i$ denote total sample number and path length, respectively. Since each viewpoint corresponds to a panorama consisting of 36 images, we ultimately obtain more than 6M visual-goal-trajectory pairs.

**Training Base Navigation Agent.** We utilize the DUET (Chen et al., 2022a), a widely used navigation model as the agent, which is pretrained from scratch and finetuned on the data $D_0$. Following (Anderson et al., 2018b), we treat navigation as a cumulative single action prediction (SAP) process. The agent takes a navigation goal, the current observation, and navigation history as input and outputs the next action decision based on the current node. It is formulated as follows:

$$\hat{\mathbf{v}} = \mathcal{N}_{\theta_0}(\mathcal{O}_i^t | \mathbf{g}_i, \mathcal{O}_i^{<t}), \tag{1}$$

where $\mathcal{N}$ is the navigation agent parametrized by $\theta_0$. $\mathcal{O}_i^t$ is the observation at the $t_{\text{th}}$ step and $\mathcal{O}_i^{<t}$ is the historical navigation records.

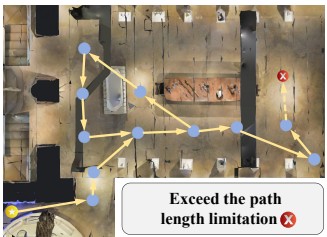
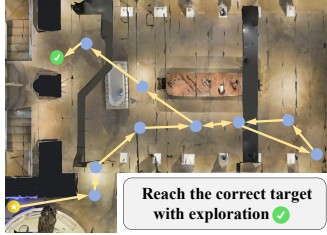

Figure 3: Filtering the trajectories generated by the navigation agent. The agent may fail in various scenarios, such as terminating at similar but incorrect targets or exceeding the path length limitation. Only the trajectories that successfully reach the correct target with efficient exploration will be retained for subsequent training iterations.

We minimize the imitation loss $\mathcal{L}$ between the agent's predicted actions toward the navigation goal and the ground-truth ones to optimize the agent $\mathcal{N}$ during finetuning, which can be formulated as:

$$\theta = \arg\min_{\theta} \left[ \sum_{i=1}^{n} \sum_{t=1}^{l_i} \mathcal{L}(\mathcal{N}_{\theta_0}(\mathcal{O}_i^t | \mathbf{g}_i, \mathcal{O}_i^{<t}), \mathbf{p}_i[t]) \right], \tag{2}$$

where $\mathbf{p}_i[t]$ denotes the ground truth location at the step $t$.

**Curating Rollout Data Generation.** After training on the shortest-path training data, the navigation agent $\mathcal{N}_{\theta_1}$ acquires fundamental exploration capabilities and is utilized to generate new training trajectories. The agent usually needs to flexibly traverse different rooms and distinguish similar but incorrect scenarios before successfully navigating to the goal viewpoint, which encapsulates the agent's exploration of the environment and can serve as the demonstrations to further train the agent. Given the visual goal $g_i$ of the target viewpoint $v_i^t$ and the starting viewpoint $v_i^s$, the agent autonomously generates step-by-step actions based on real-time observation, navigation history and the target. As shown in Figure 3, if the agent reaches the correct target within the path length limitation, the corresponding trajectory will be reserved for the next-iteration training corpus $D_1$.

**Iterative Self-Improving.** We repeat the aforementioned process leveraging the newly generated trajectories as the training set. Specifically, at the iteration $t$, we first utilize the new exploration-training data $D_{t-1}$ for pretraining and finetuning the new stronger navigation agent $\mathcal{N}_{\theta_t}$. Then we use the new agent to refine the $D_0$ to get new training data $D_t$. The reason why we choose to filter the data and train the agent from scratch rather than the previous checkpoint is that we want to eliminate the impact of knowledge in the previous low quality data. Through this self-demonstration iterative mechanism, the agent progressively enhances its exploration capabilities, ultimately outcoming a highly capable agent and high-quality dataset for subsequent utilization.

Note that during finetuning, unlike prior methods that rely solely on imitating shortest-path oracle actions via DAgger (Chen et al., 2022a), we interleave teacher forcing with demonstrations (agent strictly follows the given trajectories) and student forcing with shortest-path oracle actions (agent executes its own predicted actions but is supervised by the oracle). This alternating strategy balances exploration and exploitation, enabling both robust exploration learning and effective error correction.

## 3.2 SCALING AND CONTINUING SID IN NEW ENVIRONMENTS

Once the agent's performance converges on MP3D environments, we introduce more visual goal trajectories from HM3D (Ramakrishnan et al., 2021) environments and employ the previous agent to generate high-quality large-scale demonstrations on them. The newly generated data are integrated with the previous training data to enable the agent to simultaneously acquire novel exploration capabilities while maintaining its prior navigation knowledge. Considering the computation overhead, we continue the self-improving demonstration process for only one iteration on the whole 800 HM3D environments. Note that to verify the self-improving nature of SID after scaling up, we provide separate experiments of introducing new 60 HM3D environments beyond the original 60 MP3D scans and continuing SID for two rounds, as detailed in Sec 4.1.

Table 1: **Quantitative analysis of the computation overhead.** Caption, Feature, Trajectory and Pretraining denotes VLM-based Caption Generation, Visual Feature Extraction, Trajectory Sampling in all iterations of SID training and the final Language-Goal Pretraining, respectively.

| Stages | Caption | Feature | Trajectory | SID Training | | | | Pretraining |
|---|---|---|---|---|---|---|---|---|
| | | | | Round1 | Round2 | Round3 | Scale | |
| GPU Hours | 460 | 190 | 1500 | 140 | 180 | 260 | 1200 | 1500 |

### 3.3 Trajectory Transfer to Goal-Oriented Language-Guided Navigation

As depicted in Figure 2, the enhanced agent generates more efficient exploration trajectories, which is then transferred with VLM caption augmentation to align with daily life applications and high-level VLN tasks like SOON (Zhu et al., 2021) and REVERIE (Qi et al., 2020). Specifically, we leverage InternVL2-26B (Chen et al., 2024) to generate semantically rich image descriptions that capture object localization, environmental context and spatial relationships. To maximize data scale while minimizing the redundancy caused by the overlap of different images, we interleavedly choose 18 out of 36 images from the panorama, resulting in 46M high-quality language goal exploration trajectories.

**Comparison to Previous VLN datasets.**
Table 2 presents detailed statistics of our and previous VLN datasets. As afore-mentioned, the lack of demonstrations on exploration strategies hinders the advancement of goal-oriented navigation. Existing methods mainly concentrate on generating instructions to tackle data scarcity or improve instruction quality. In contrast, SID focuses on the generation of robust and diverse trajectories.

Table 2: **Statistics of different VLN training data.**

| Dataset | Generated | Demonstration | #Env. | #Data |
|---|---|---|---|---|
| *Fine-grained VLN* | | | | |
| R2R (Anderson et al., 2018b) | ✗ | - | 61 | 14,039 |
| RxR-en (Ku et al., 2020) | ✗ | - | 60 | 26,464 |
| Marky (Wang et al., 2022) | ✓ | - | 60 | 333,777 |
| *Goal-oriented VLN* | | | | |
| REVERIE (Qi et al., 2020) | ✗ | ✗ | 60 | 10,466 |
| SOON (Zhu et al., 2021) | ✗ | ✗ | 34 | 27,800 |
| AutoVLN (Chen et al., 2022b) | ✓ | ✗ | 900 | 217,703 |
| ScaleVLN-RVR (Wang et al., 2023b) | ✓ | ✗ | 1289 | 831,318 |
| NavRAG (Wang et al., 2025b) | ✓ | ✗ | 861 | 2,115,019 |
| SID-46M (Ours) | ✓ | ✓ | 860 | 46,534,355 |

What evolves across different iterations in our method is not the instructions but the navigation paths, which become increasingly exploratory and efficient through self-improvement. Consequently, the navigation agent can effectively learn exploration strategies from its own trajectories, reduce its error rate, and navigate to the final goal with greater efficiency. In total, our dataset contains 46 million language goal exploration trajectories with an average of 9.01 viewpoints, which is the first to provide large-scale demonstrations on exploration strategies in goal-oriented VLN.

## 4 Experiments

**Implementation Details.** We utilize DINOv2 (Oquab et al., 2023) and SigLIP (Zhai et al., 2023) as the visual encoders in our experiments following OpenVLA (Kim et al., 2024). Furthermore, we utilize LMDeploy to deploy InternVL2-26B and annotate a total of 1,213,479 images. We conduct three iterative rounds of SID training on MP3D environments and one iteration of scaling up on 800 HM3D environments. Finally, we use the weights of LXMERT (Tan & Bansal, 2019) to initialize DUET, pretrain the agent from scratch for 124k iterations with a batch size of 1024 at a learning rate of $1 \times 10^{-5}$ on 8 GPUs, and finetune it on the downstream tasks. The detailed analysis of the computational cost is presented in Table 1, with all experiments conducted on NVIDIA A800 GPUs.

**Datasets and Metrics.** We perform experiments on goal-oriented navigation tasks, including high-level vision-language navigation( REVERIE (Qi et al., 2020), SOON (Zhu et al., 2021)) and Object Goal Navigation (Savva et al., 2019). The datasets are outlined as follows:

- **REVERIE** consists of a vocabulary of over 1,600 words and 21,702 human-annotated navigational instructions, each describing a target object within MP3D environments (Chang et al., 2017). The average length of the collected instructions is 18 words, involving both navigation and referring expression. The shortest path from the agent's initial location to the target location is between 4 to 7 steps, and the agent needs to find and localize the object in the observation.

- **SOON** includes 3,848 instructions with 1,649-word vocabulary, featuring trajectories in 38 MP3D houses. It also follows the same train/val/test split strategy as REVERIE. The instructions' average length reaches 38.6 words, with trajectories spanning 15-60+ meters and surpassing REVERIE's paths. Due to the increased task difficulty and smaller training dataset size, it is more challenging to achieve high performance on SOON than on REVERIE.

Table 3: **Detailed Analysis of multi-round SID Learning.** Left: Performance on Image Goal Navigation validation unseen split and statistics of the training trajectories data. Right: Performance in unseen environments on REVERIE and SOON. # VP denotes the number of viewpoints.

| Round | Image Goal Navigation Training | | | | | Transferred VLN Performance | | | |
| | Val Unseen | | | Traj Statistics | | REVERIE | | SOON | |
| | OSR↑ | SR↑ | SPL↑ | Traj Num | Avg #VP | SR↑ | SPL↑ | SR↑ | SPL↑ |
|---|---|---|---|---|---|---|---|---|---|
| *SID in MP3D environments* | | | | | | | | | |
| 1 | 61.85 | 61.45 | 40.24 | 5,082,528 | 8.27 | 50.87 | 36.56 | 45.02 | 30.58 |
| 2 | 65.50 | 65.35 | 43.31 | 5,698,944 | 7.31 | - | - | - | - |
| 3 | 65.55 | 65.45 | 44.62 | 5,873,474 | 7.27 | 51.92 | 39.80 | 46.46 | 34.51 |
| *Scaling up with 800 HM3D environments* | | | | | | | | | |
| **4** | **75.00** | **75.00** | **54.67** | 93,069,068 | 9.01 | **59.39** | **45.95** | **50.88** | **38.42** |

Table 4: **Performance on Image Goal Navigation when continuing SID with 60 additional HM3D environments for two rounds.**

| Tot Env | Validation Seen | | | Validation Unseen | | |
| | OSR↑ | SR↑ | SPL↑ | OSR↑ | SR↑ | SPL↑ |
|---|---|---|---|---|---|---|
| 60 | 87.40 | 86.60 | 79.18 | 65.55 | 65.45 | 44.62 |
| 120 (Round 1) | 86.80 | 86.80 | 78.38 | 70.00 | 69.90 | 48.58 |
| **120 (Round 2)** | **89.20** | **88.60** | **79.24** | **70.80** | **70.75** | **48.60** |

Table 5: **Statistics of room exploration of the trajectories on the image goal navigation validation unseen split.**

| Trajectory Source | Room Statistics | | |
| | Num | Type | Target |
|---|---|---|---|
| Original Shortest Paths | 2.77 | 2.62 | 1.09 |
| Agent trained on MP3D | 4.21 | 3.05 | 1.38 |
| Agent scaled on HM3D | 3.89 | 2.94 | 1.32 |

- **Object Goal Navigation** requires an agent to navigate to the target specified by a general object category. To verify the generalization of SID to object goal navigation, we follow SAME (Zhou et al., 2024b) to transfer the validation split of the Habitat ObjectNav dataset (Savva et al., 2019) to evaluate agent performance in discrete environments, as detailed in Section 4.3.

We evaluate our agent using standard path-fidelity metrics, including Trajectory length (TL), Success Rate (SR), Oracle Success Rate (OSR), Success Rate Weighted by Path Length (SPL) (Anderson et al., 2018a). Following (Qi et al., 2020), we adopt Remote Grounding Success (RGS) and RGS weighted by Path Length (RGSPL) to evaluate object grounding. The agent is considered successful only if it stops accurately at the designated target viewpoint in image-goal navigation.

## 4.1 SELF-IMPROVING DEMONSTRATION RUNNING RESULTS

**Consistent Self improvement of navigation agent.** SID generates both novel effective trajectories and enhanced agents from the given environments via self-improving demonstrations. As shown in the left part of Table 3, the SPL on unseen environments improves from 40.24% to 44.62%. And the close alignment of OSR and SR indicates the agent possesses a robust capability to identify the target— once it traverses the goal location, it can accurately terminate the episode there. Furthermore, the successfully sampled trajectories by the agent increase while the average length of them continuously decreases from 8.27 to 7.27 concurrently. This highlights the self-bootstrapping nature of SID: the agent iteratively learns better exploration strategies through its own demonstrations.

**Scalability and suitability to new environments.** When SID converges in MP3D environments, we show it can scale up and continue to self-improve when extended to new environments. (1) Additional data from 800 HM3D environments yields a clear 10% improvement in SR and SPL in unseen environments as shown in Table 3. (2) Scaling up SID to 60 additional HM3D environments and continuing two rounds of processes shows that SID retains its self-improving nature with increased data in Table 4, demonstrating its effective scalability with larger training environments.

**Transferability to language-guided navigation.** Table 3 reports the performance improvement gained through iterative self-demonstrations on SOON and REVERIE benchmarks with transferred language-goal navigation training. These results indicate the strong generalizability and transferability of SID exploration trajectories to goal-oriented VLN tasks.

Table 6: **Comparison of different strategies of supervision in the teacher forcing.** Left: Performance on image goal navigation with MP3D data finetuning. Right: Performance on REVERIE validation unseen split with large-scale SID data augmentation.

| Strategies | Image Goal Navigation Finetuning | | | | | | Augmented REVERIE Finetuning | | | | | |
|---|---|---|---|---|---|---|---|---|---|---|---|---|
| | Validation Seen | | | Validation Unseen | | | Navigation | | | | Grounding | |
| | OSR↑ | SR↑ | SPL↑ | OSR↑ | SR↑ | SPL↑ | TL↓ | OSR↑ | SR↑ | SPL↑ | RGS↑ | RGSPL↑ |
| Shortest | 81.80 | 80.40 | 68.32 | 65.05 | 64.90 | **44.63** | 20.95 | **63.62** | 58.53 | 44.04 | 37.18 | 27.93 |
| **Explored** | **83.00** | **82.40** | **71.40** | **65.50** | **65.35** | 43.31 | 20.78 | 63.45 | **59.39** | **45.95** | **38.54** | **29.69** |

Table 7: **Comparison on VLN validation unseen splits with different demonstration sources.**

| Sources | REVERIE | | | SOON | | |
|---|---|---|---|---|---|---|
| | OSR↑ | SR↑ | SPL↑ | OSR↑ | SR↑ | SPL↑ |
| Shortest | 55.10 | 50.87 | 36.56 | 54.49 | 46.02 | 30.58 |
| Random | 54.22 | 50.87 | 37.04 | 52.36 | 42.18 | 32.48 |
| **Agent** | **56.09** | **51.92** | **39.80** | **55.28** | **46.46** | **34.34** |

Table 8: **Comparison on VLN validation unseen splits with different scaling strategies.**

| Strategies | REVERIE | | | SOON | | |
|---|---|---|---|---|---|---|
| | OSR↑ | SR↑ | SPL↑ | OSR↑ | SR↑ | SPL↑ |
| *w/o* | 56.09 | 51.92 | 39.80 | 55.28 | 46.46 | 34.51 |
| Shortest | 60.97 | 55.16 | 42.58 | 56.43 | 48.35 | 37.65 |
| **Explored** | **63.45** | **59.39** | **45.95** | **60.32** | **50.88** | **38.42** |

**Room Exploration.**    Regarding our navigation tasks are performed in 3D indoor environments consisting of interconnected rooms, evaluating the room exploration and error correction is crucial. As shown in Table 5, the agent trained on self-exploration data explores a greater number and variety of rooms than the shortest paths. Besides, the number of rooms explored of the same type as the target has increased, verifying that the agent actively explores similar rooms and corrects its wrong steps to reach the correct destination. The navigation efficiency and success rate continuously improve with the progress of SID, showcasing that the agent effectively learns exploration strategies from its own trajectories, reduces its error rate, and navigates to the final goal with greater efficiency. We provide a case of step-by-step room exploration and trajectory evolution in Section D in the Appendix.

## 4.2    ABLATION AND ANALYSIS

**Supervision Strategies.**    We compare different supervision strategies in the teacher forcing in Table 6. (1) Self-demonstrations benefits finetuning across different data scales and goal modals, revealing the significance of learning from self-exploration strategies instead of greedily taking the shortest paths. (2) Even with the same shortest-path demonstrations for finetuning, the image-goal navigation agent pretrained with self-demonstrations (Table 6, line 1) still performs significantly better than the one pretrained with shortest-path demonstrations (Table 3, line 1). This further underscores that such self-demonstrations benefit both pretraining and finetuning.

**Demonstration Sources.**    We show that the agent gains increase from self-demonstrations rather than random exploration. We sample random paths for goal-oriented navigation data. The number of viewpoints of these trajectories ranges from 7 to 11 and the average number is 8.9, similar to the distribution of the agent's exploration ones. Table 7 shows that the agent trained on the randomly sampled trajectories can only reach the equivalent performance with the shortest path but far behind the exploration paths of the agent itself, which strongly claims that the agent learns effectively from its own exploration rather than arbitrary trajectories, validating SID for goal-oriented navigation.

**Effectiveness of SID beyond Simple Data Scaling.**    To verify the effect of the agent's demonstrations in new environment rather than simply scaling up, we conduct another comparison by merely adding shortest-path trajectories in HM3D environments to train a new agent. The results in Table 8 shows that incorporating shortest path data modestly improves performance but is distinctly surpassed compared to leveraging the agent's exploration trajectories, which demonstrates that the primary benefit arises from the SID paradigm rather than simply introducing more data in new environments.

**Limitation.**    SID's exploration can fail in some intricate scenarios. If the agent continuously explores the environment, it will exceed the maximum navigation steps limitation and be forced to stop. Furthermore, SID agent is pretrained on robotic navigation assumptions, which may need additional adaptation for practical applications. And the successful transfer of SID to VLN tasks relies on captions generated by VLMs, which may be incomplete or inaccurate due to hallucinations. We present the detailed analysis in Section E in the Appendix.

Table 9: **Comparison with the state-of-the-art methods on SOON datasets.**

| Methods | Validation Unseen | | | | | Test Unseen | | | | |
|---|---|---|---|---|---|---|---|---|---|---|
| | TL↓ | OSR↑ | SR↑ | SPL↑ | RGSPL↑ | TL↓ | OSR↑ | SR↑ | SPL↑ | RGSPL↑ |
| Human | - | - | - | - | - | - | 91.4 | 90.4 | 59.2 | 51.1 |
| DUET (Chen et al., 2022a) | 36.2 | 50.9 | 36.3 | 22.6 | 3.8 | 41.8 | 43.0 | 33.4 | 21.4 | 4.2 |
| NaviLLM (Zheng et al., 2024) | - | - | 38.3 | 29.2 | - | - | - | 35.0 | 26.3 | - |
| KERM (Li et al., 2023) | 35.8 | 51.6 | 38.1 | 23.2 | 4.0 | - | - | - | - | - |
| GridMM (Wang et al., 2023a) | 38.9 | 53.4 | 37.5 | 24.8 | 3.9 | 46.2 | 48.0 | 36.3 | 21.3 | 4.2 |
| MBA (Zhang et al., 2024a) | 37.2 | - | 42.0 | 29.6 | 6.1 | 36.5 | - | 38.8 | 26.2 | 6.3 |
| GOAT (Wang et al., 2024a) | - | 54.7 | 40.4 | 28.1 | 5.1 | - | 50.6 | 40.5 | 25.2 | 6.1 |
| AutoVLN (Chen et al., 2022b) | - | 53.2 | 41.0 | 30.7 | 4.1 | - | 48.7 | 40.4 | 27.8 | 5.1 |
| Meta-Explore (Hwang et al., 2023) | - | 52.7 | 44.7 | 34.8 | **8.9** | - | 48.7 | 39.1 | 25.8 | 4.0 |
| **SID (Ours)** | **33.3** | **60.3** | **50.9** | **38.4** | 6.9 | **36.4** | **54.0** | **47.7** | **35.2** | **8.3** |

Table 10: **Comparison with the state-of-the-art methods on REVERIE datasets.** *-SD denotes the same method with SigLIP and DinoV2 as vision encoders for fair comparison.

| Methods | Validation Unseen | | | | | | Test Unseen | | | | | |
|---|---|---|---|---|---|---|---|---|---|---|---|---|
| | TL↓ | OSR↑ | SR↑ | SPL↑ | RGS↑ | RGSPL↑ | TL↓ | OSR↑ | SR↑ | SPL↑ | RGS↑ | RGSPL↑ |
| Human | - | - | - | - | - | - | 21.2 | 86.8 | 81.5 | 53.7 | 77.8 | - |
| DUET (Chen et al., 2022a) | 22.1 | 51.1 | 47.0 | 33.7 | 32.1 | 23.0 | 21.3 | 56.9 | 52.5 | 36.1 | 31.9 | 22.1 |
| NaviLLM (Zheng et al., 2024) | - | 53.7 | 44.6 | 36.6 | - | - | - | 56.2 | 43.5 | 34.5 | - | - |
| BEVBert (An et al., 2023) | - | 56.4 | 51.8 | 36.4 | 34.7 | 24.4 | - | 57.3 | 52.8 | 36.4 | 32.1 | 22.1 |
| BSG (Liu et al., 2023) | 24.7 | 58.1 | 52.1 | 35.6 | 35.4 | 24.2 | 22.9 | 62.8 | 56.5 | 38.7 | 33.2 | 22.3 |
| VER (Liu et al., 2024) | 23.0 | 61.1 | 56.0 | 39.7 | 33.7 | 23.7 | 24.7 | 62.2 | 56.8 | 38.8 | 33.9 | 23.2 |
| GOAT (Wang et al., 2024a) | - | - | 53.4 | 36.7 | 38.4 | 26.1 | - | - | 57.7 | 40.5 | **38.3** | 26.7 |
| *Paradigm with the same 800 environments from HM3D* | | | | | | | | | | | | |
| AutoVLN (Chen et al., 2022b) | - | 62.1 | 55.9 | 40.9 | 36.6 | 26.8 | - | 62.3 | 55.2 | 38.9 | 32.2 | 22.7 |
| ScaleVLN (Wang et al., 2023b) | - | 63.9 | 57.0 | 41.8 | 35.8 | 26.1 | - | 62.7 | 56.1 | 39.5 | 32.5 | 22.8 |
| NavRAG (Wang et al., 2025b) | - | **70.7** | 57.3 | 42.0 | - | - | - | - | - | - | - | - |
| ScaleVLN-SD (Wang et al., 2023b) | - | 64.8 | 58.4 | 43.5 | 37.7 | 28.1 | - | - | - | - | - | - |
| **SID (Ours)** | **20.8** | 63.5 | **59.4** | **46.0** | **38.5** | **29.7** | **20.2** | **65.5** | **60.4** | **45.9** | 36.6 | **27.7** |

### 4.3 COMPARISONS ON GOAL-ORIENTED LANGUAGE-GUIDED NAVIGATION TASKS

**SOON and REVERIE.** As shown in Table 9, SID reaches new state-of-the-art results across all navigation metrics on the challenging SOON task, exceeding AutoVLN with fewer training 3D environments. Notably, on the SOON test-unseen leaderboard, SID outperforms all previous methods and achieves the highest success rate. On the REVERIE task, Table 10 shows that SID also achieves new state-of-the-art performance, which underscores the significance of efficient demonstrations on exploration strategies instead of simple instruction augmentation for challenging goal-oriented VLN. Furthermore, the trajectory length of SID proves that our enhancement stems from genuinely more efficient exploration routes rather than merely allowing successful yet longer episodes.

**Object Goal Navigation.** As the Table 11 shows, SID notably achieves a 76% success rate on the transferred objectnav validation unseen set, surpassing DUET by a large relative margin of 8%. This indicates that the exploration capabilities learned by SID from self-demonstrations can effectively generalize to object-goal navigation. The experimental details are presented in Section C in the Appendix.

Table 11: **Comparison on the Transferred ObjectNav MP3D Validation split.**

| Agent | ObjectNav-MP3D(Val) | | | |
|---|---|---|---|---|
| | TL↓ | NE↓ | SR↑ | SPL↑ |
| DUET | **22.17** | 3.67 | 68 | 29 |
| **SID** | 24.12 | **2.74** | **76** | **37** |

## 5 CONCLUSION

In this paper, we introduce SID for learning goal-oriented navigation with Self-Improving Demonstrations at scale. SID is a scalable and transferrable learning paradigm that enables agents to iteratively enhance their navigation capabilities through self-generated, exploration-rich trajectories. Leveraging agent exploration as supervision, SID mitigates the reliance on costly human annotations for large-scale demonstrations. Our experiments indicate a consistent improvement of SID in the classical image-goal navigation task. Additionally, we show that SID trajectories can be seamlessly transformed into language-guided navigation with VLM caption augmentation and elevate the new state-of-the-art performance across diverse goal-oriented VLN tasks, demonstrating great potential for unified goal-oriented navigation. In general, SID sets a new standard for scalable goal-oriented navigation learning and opens promising directions for self-improving embodied learning.

## ETHICS STATEMENT

I acknowledge that I and all co-authors of this work have read and commit to adhering to the ICLR Code of Ethics throughout the research and submission process. This work does not involve human subjects, human data, or animal experiments. The datasets used in our experiments are either publicly available benchmark datasets or synthetically generated for research purposes, and we do not release any new datasets that contain personally identifiable or sensitive information. Our methodology and findings are intended to advance the scientific understanding of machine learning models and do not pose direct risks of misuse or harm. We are committed to fairness, transparency, and accountability in research, and we encourage responsible use of the techniques described in this paper.

## REPRODUCIBILITY STATEMENT

To ensure the reproducibility of our results, we have taken several steps. Hyperparameters and training configurations are fully documented in Section 4 and further detailed in Section C in the Appendix. The datasets used are publicly available and properly cited. We also include detailed preprocessing pipelines and mathematical derivations supporting our theoretical claims. We are currently organizing the code and data and promise to release it on GitHub in the future.

## THE USE OF LARGE LANGUAGE MODELS

We only use LLMs to aid or polish writing. We also utilize VLM, specifically InternVL2-26B (Chen et al., 2024) for data annotation. The details are described in the paper.

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

APPENDIX

We first describe the implementation details of our experiments in Section A, including pretraining objectives and details of Self-Improving Demonstrations experiments. In Section B, we provide detailed experiments about transferring SID to language-guided tasks and the effects of caption styles. Section C presents the experimental details of downstream navigation tasks. Section D visualizes our navigation trajectories towards the target. Section E discusses some limitations of our SID method.

## A  IMPLEMENTATION DETAILS

### A.1  NAVIGATION AGENT

We employ the DUal-scale Graph Transformer (DUET) (Chen et al., 2022a) a widely used navigation model as the agent. The core innovation of DUET is its dual-scale reasoning mechanism fusing information from coarse and fine-grained encoders. The coarse-grained global encoder operating on the global topological map leverages a graph-aware self-attention mechanism to effectively incorporate the map's structure, enabling robust reasoning over extended distances and complex spatial layouts. While the fine-grained local encoder processes detailed local visual information from the current viewpoint (including panoramic imagery and object features) for precise language grounding and local action choices. This combination allows DUET to effectively balance nuanced instruction comprehension with strategic long-range planning.

### A.2  PRETRAINING OBJECTIVES

We mainly employ two proxy tasks, MLM and SAP, to pretrain the agent. Here we describe these two proxy tasks in detail. The inputs for these tasks are goal-trajectory pairs as $D = \{(\mathbf{p}_i, \mathbf{g}_i,)\}_{i=1,...,n}$, where $\mathbf{p}_i = <\mathbf{v}_i^j>_{j=1,...l_i}$ is the demonstration trajectories reaching the goal $\mathbf{g}_i$ and $\mathbf{g}_i$ is the description (*e.g.*, an image or a sentence capturing the destination) of the target location $\mathbf{p}_i[-1]$ (the end location of the path $\mathbf{p}_i$). $n$ and $l_i$ denote total sample number and path length, respectively. Among them, SAP is utilized throughout the multi round self-improving demonstrations, while MLM is adopted only during trasfering and we sample one task for each iteration with equal probability.

**Masked Language Modeling (MLM)**    involves predicting masked words based on textual context and the full trajectory. In this task, the description $\mathbf{g}_i$ of the target location is equivalent to instruction $\mathcal{W} = \{w_1, w_2, \ldots, w_n\}$. A special [mask] token is used to randomly mask out 15% of the tokens in $\mathcal{W}$. We predict the masked word distribution $p(w_i|\hat{\mathcal{W}}, \mathcal{P}) = f_{\mathrm{MLM}}(x_i)$ through a two-layer fully-connected network, where $\hat{\mathcal{W}}$ is the masked instruction, $x_i$ is the output embedding of the masked word $w_i$ and $\mathcal{P}$ is the corresponding trajectory. The objective is to minimize the negative log-likelihood of predicting the original words: $\mathcal{L}_{\mathrm{MLM}} = -\log p(w_i|\hat{\mathcal{W}}, \mathcal{P})$.

**Single Action Prediction (SAP)**    aims to predict the next action based on the goal description and the given path. Following (Chen et al., 2022a), we predict the probability for each candidate action in the action space via a two-layer fully-connected network. The objective is to minimize the negative log probability of the target view action $\mathcal{L}_{\mathrm{SAP}} = -\log p_t(a_t|\mathcal{W}, \mathcal{P}_{<t})$.

Specifically, we refine the sampling strategy for selecting viewpoints during training to enhance both path adherence and error correction. The original probability distribution in DUET is 20% for the goal viewpoint (to learn stopping), 40% for other on-path viewpoints (to learn path progression), and 40% for random off-path viewpoints (to learn error correction). However, considering that on-path points constitute only a very small fraction of the total navigable viewpoints within an environment, random sampling of off-path points provides sparse and often uninformative negative examples for effective error correction. Furthermore, our finetuned models demonstrate a high Oracle Success Rate (OSR) that closely aligns with their Success Rate (SR), which indicates that the agent is highly proficient at making the decision to stop if it reaches the correct final viewpoint.

Based on these observations, we introduce a revised sampling strategy. We allocate 75% of our sampling probability to uniformly selecting points along the self-demonstration trajectories, on which we want the agent to focus more. The remaining 25% of the sampling probability is dedicated to

selecting challenging negative examples specifically from the candidate navigable viewpoints. This targeted negative sampling strategy is informed by an analysis of agent failures on 2000 unseen trajectories. We observed a distinct error distribution across trajectory steps (*e.g.*, [927, 295, 253, 189, 214, 98, 24] errors at steps 1 to N, respectively). Therefore, our revised strategy prioritizes sampling difficult negatives (*i.e.*, off-path candidate viewpoints where the agent is likely to err), providing more effective supervision for learning robust error correction and improving overall navigation policy.

### A.3 DETAILS OF EXPERIMENTS OF SELF-IMPROVING DEMONSTRATION

Goal-oriented VLN contains trajectories that lead to target objects specified by high-level instructions. Following their distribution of the data, for every visible object at a viewpoint, we sample paths with a length between 5 and 7 vps that end at the viewpoint, resulting in 181,246 paths from MP3D environments. Since each viewpoint corresponds to a panorama consisting of 36 images, we ultimately obtain 6,524,856 visual-goal trajectories. Then we utilize the weights of LXMERT (Tan & Bansal, 2019) to initialize DUET and pretrain the model with our synthetic visual-goal datasets.

We utilize DINOv2-Base (Oquab et al., 2023) and SigLIP-Base (Zhai et al., 2023) as the visual encoders to extract the image features. For the first round, we pretrain DUET for 200k iterations with a batch size of 128 and a learning rate of $5 \times 10^{-5}$ on MP3D environments. We compare the model checkpoints and pick the one with the highest finetuning performance. Then we finetune DUET for 300k iterations, with batch size 64 and learning rate $5 \times 10^{-5}$ on 8 NVIDIA A800 GPUs.

We repeat the aforementioned process for three rounds to get a strong navigation agent, which is then adopted to generate new trajectories for the visual goals in the original MP3D and scale up in new 800 HM3D environments. We follow the same sampling rule as in MP3D environments, resulting in 3220,504 paths and 115,938,144 visual-goal-trajectory pairs from HM3D. Then we use the navigation agent trained through self-improving demonstrations to generate new paths and obtain totally 60M visual-goal trajectories from 860 MP3D and HM3D environments. Based on the new training corpus, we continue the above process SID for another round and get the final self-demonstration trajectories.

## B TRANSFERRING TO LANGUAGE-GUIDED TASKS

To transfer the trajectories with demonstration to language-guided navigation tasks, we leverage VLM to generate captions for the images of the target. Specifically, to maximize data scale and minimize the redundancy caused by the overlap of different images, we choose 18 out of 36 images from the panorama in a interleaved way. We first employ InternVL2-26B (Zhu et al., 2025) to generate three styles of caption on MP3D environments to explore the most suitable style for caption augmentation:

- Detail-style: This prompt generates detailed descriptions with object localization (*e.g.*, "to the left of the door"), environmental context (*e.g.*, "the staircase connects the living room") and positional relation(*e.g.*, "the left/right/above"), providing the navigator fine-grained visual information, with the template shown in Figure 4.

- REVERIE-style: This prompt generates REVERIE-like annotations, using linguistic phenomena such as dangling modifiers and spatial relations and focuses on concise navigation instructions for Remote Embodied Visual Referring Expression in Real Indoor Environments (Qi et al., 2020), with the template shown in Figure 5.

- SOON-style: This prompt produces annotations with object names, attributes (*e.g.*, "cylindrical, metallic") and relationships as well as target and neighbor areas(*e.g.*, "next to the dining room"), strictly following the setting of Scenario Oriented Object Navigation(SOON) (Zhu et al., 2021), with the template shown in Figure 6.

After getting three styles of caption, we transfer the training data by replacing the images with corresponding captions but keeping the trajectories created by the navigation agent, resulting in 3M caption-trajectories training corpus. Although the REVERIE-Style and SOON-Style captions are more similar to the instructions in the downstream VLN tasks, the Detail-Style captions lead to the best zero-shot and finetune performance on both REVERIE and SOON as shown in Table 13. The agent pretrained on detail-style caption-trajectory pairs surpasses the other ones after being finetuned on the downstream tasks and the baseline model (pretrained on initial, non-caption data) on SOON.

**[Prompt]**
Image-1: <image> Image-2: <image>
Task Overview: You will now input two images at a time: the first image is the current viewpoint panorama, which provides background context, and the second image is the one that requires annotation. Your task is to generate a detailed description for the second image.
Description Requirements:
1. Object Localization:
- Each description should identify specific objects in the second image (e.g., switches, doors, furniture).
- Mention the location of these objects and describe their position relative to other elements in the environment (e.g., "to the left of the door").
- Describe the environmental context, including surrounding rooms, corridors, or furniture (e.g., "the staircase connects the living room and kitchen").
2. Conciseness: Each description should be clear and concise, with a maximum of 40 words.
3. Template: Each description should starts with This image shows... Notes: Be careful about the positional relation, especially the information about the right/left/above etc. Positional Information: This is the agent's downward/eye-level/upward point of view.

**[Model Output]**
This image shows a room with a large round table in the foreground, featuring a blue and white vase. To the right, there is a wooden door with glass panels leading to another room. A chandelier hangs from the ceiling, and a chair is positioned near the window. The room has dark wooden paneling and a rug on the floor.

Figure 4: Prompt and Model Output of the Detail-style Captions.

**[Prompt]**
Image-1:<image> Image-2:<image>

Task Overview:
The REVERIE task requires an intelligent agent to correctly localize a remote target object (which cannot be observed at the starting location) specified by a concise high-level natural language instruction. Since the target object is in a different room from the starting one,the agent needs first to navigate to the goal location.
You will now input two images at a time:the first image is the current viewpoint panorama, which provides background context, and the second image is the one that requires annotation. Your task is to generate an annotation for the second image that matches the style of the provided REVERIE examples.

Annotation Requirements:
1.Linguistic Phenomena:
-The annotation should include various linguistic phenomena such as:
-Dangling Modifiers(e.g.,"Fold the towel in the bathroom with the fishing theme").
-Spatial Relations(e.g.,"Go to the blue family room and bring the framed picture of a person on a horse at the top left corner above the TV").
-Imperatives(e.g.,"Go to the hallway area where there are three pictures side by side and get me the one on the right").
-Co-references(e.g.,"There is a bottle in the office alcove next to the piano.It is on the shelf above the sink on the extreme right.Please bring it here").
2.Conciseness:
-Each annotation should be clear and concise,with a maximum of 40 words.
-Avoid unnecessary details while ensuring the instruction is sufficient for localization.
3.Navigation Focus:
-The agent is required to identify the referent object but is not required to complete any manipulation tasks(e.g.,folding the towel,dusting the mirror).
-Focus on providing clear navigation instructions to locate the target object.

Example Annotations:
1.Enter the bedroom with the letter E over the bed and turn the light switch off.
2.Windex the mirror above the sink, in the bedroom with the large,stone fireplace.
3.Could you please dust the light above the toilet in the bathroom that is near the entry way?
4.At the top of the stairs, the first set of potted flowers in front of the stairs need to be dusted off.
5.To the right at the end of the hall, where the large blue table foot stool is, there is a mirror that needs to be wiped.
Positional Information: This is the agent's downward/eye-level/upward point of view.

**[Model Output]**
Go to the room with the large wooden beams and high ceiling. There is a cabinet with glass doors on the left side of the room.

Figure 5: Prompt and Model Output of the REVERIE-style Captions.

**[Prompt]**
Image-1:<image>Image-2:<image>

Task Overview:
The Scenario Oriented Object Navigation(SOON) task requires an intelligent agent to navigate from an arbitrary position in a 3D embodied environment to localize a target object following an instruction. The task includes two sub-tasks: navigation and localization.
You will now input two images at a time:the first image is the current viewpoint panorama, which provides background context, and the second image is the one that requires annotation. Your task is to generate an annotation for the second image that matches the style of the provided SOON examples.

Annotation Requirements:
1.Object Description:
-Describe the object name,attributes,and relationships with other objects.
-Mention the target area and neighbor areas to provide clear navigation instructions.
2.Linguistic Structure:
-Follow the six-step annotation process as shown in the example:
-Object Name: Identify the target object (e.g.,"lamp").
-Object Attribute: Describe its attributes (e.g.,"cylindrical,metallic,tall").
-Object Relationship: Describe its relationship with other objects (e.g.,"on the cabinet").
-Target Area: Specify the room or area where the object is located (e.g.,"bright living room").
-Neighbor Areas: Mention nearby rooms or areas (e.g.,"next to the dining room").
-Full Instruction: Combine all elements into a complete instruction.
3.Conciseness:
-Each annotation should be clear and concise, with a maximum of 40 words.
-Ensure the instruction is sufficient for the agent to locate the target object.

Example Annotation:
I want to find a cylindrical,metallic and tall lamp, which is set in the bright living room. The lamp is on the cabinet which is on the left of the television and next to the window. The living room is on the first floor, next to the dining room and next to the kitchen.
Positional Information: This is the agent's downward/eye-level/upward point of view.

**[Model Output]**
I want to find a large, stone fireplace, which is set in the cozy living room. The fireplace is on the right side of the room, next to the wooden railing and below the moose head. The living room is on the first floor, next to the dining area and near the staircase.

Figure 6: Prompt and Model Output of the SOON-style Captions.

It also reaches comparable performance on REVERIE. And the competitive zero-shot performance also demonstrates the navigation's capabilities of visual-language-action alignment gained from Detail-style captions pretraining.

With the empirical results above, we finally generate large-scale detailed-style caption on HM3D environments and leverage the transferred detail-caption training data to pretrain DUET for 124k iterations with a batch size of 1024 and a learning rate of $1 \times 10^{-5}$ with SAP and MLM. The pretrained agent is then utilized to finetune on SOON, REVERIE and Object Goal Navigation.

Table 12: **Zero-shot performance** in unseen environments on REVERIE with different styles of pretraining data.

| Styles | REVERIE | | | | |
| --- | --- | --- | --- | --- | --- |
| | OSR↑ | SR↑ | SPL↑ | RGS↑ | RGSPL↑ |
| REVERIE | 38.57 | 24.03 | 15.43 | 4.23 | 2.79 |
| SOON | 41.86 | 27.26 | 14.67 | 4.74 | 2.47 |
| **Detail** | **52.88** | **29.59** | **16.94** | **5.54** | **3.59** |

Table 13: **Finetune performance** in unseen environments on REVERIE and SOON with different styles of pretraining data.

| Styles | REVERIE | | | SOON | | |
| --- | --- | --- | --- | --- | --- | --- |
| | OSR↑ | SR↑ | SPL↑ | OSR↑ | SR↑ | SPL↑ |
| DUET | 53.59 | 50.58 | 38.27 | 54.57 | 39.23 | 26.99 |
| SOON | 52.88 | 47.63 | 34.69 | 48.82 | 39.53 | 26.29 |
| REVERIE | 50.36 | 44.33 | 32.45 | 48.67 | 37.91 | 28.16 |
| **Detail** | **55.10** | **50.87** | **36.56** | **54.49** | **46.02** | **30.58** |

## C    EXPERIMENTAL DETAILS OF THE DOWNSTREAM NAVIGATION TASKS.

Here we provide experimental details on downstream tasks, with the results presented in the main paper. For REVERIE, we finetune the DUET model with a batch size of 16, learning rate of $3 \times 10^{-5}$ and ml_weight of 0.1 with our SID data augmentation for 20k iterations. On SOON, we use a batch size of 8, learning rate of $3 \times 10^{-5}$ and ml_weight of 0.2 without augmentation for 5k iterations.

For Object Goal Navigation, we follow SAME (Zhou et al., 2024b) to adapt the Habitat-MP3D dataset (Savva et al., 2019) by discretizing its environment into a connectivity graph G. Trajectories from Habitat-Web are then matched to the nearest nodes on G based on Euclidean distance, with sequentially repeated nodes merged, resulting in 58,803 trajectories with an average of 20 steps. We similarly transfer data from the MP3D validation split to evaluate model performance in these discrete environments. After pretraining the DUET with the transferred data, we finetune it and the SID agent on this Object Goal Navigation task, as presented in Section 4.3.

## D    VISUALIZATION OF THE TRAJECTORIES

Figure 7 visualize the agent's exploration trajectories and the predicted actions at each step during multi-round self-improving. At the first round, the navigation agent learns from the sampled shortest-path trajectory. But the capability gained cannot help the agent to distinguish between two adjacent highly-similar rooms. So in the next round, the agent actively explores the two rooms and finds out the first room lacks critical elements(*e.g.*, the red valance and the computer on the desk to the left). Then it returns and explores the other room, correctly reaching the target location. The navigation agent learns from its own demonstration on the exploration strategies and picks the right room directly in the third round, which reveals the agent's capability of fine-grained visual grounding and trajectory optimization as well as the advantage of SID in sophisticated goal-oriented navigation tasks.

## E    LIMITATION

SID's exploration can fail in complex areas with numerous choices at critical navigational viewpoints. If the agent continuously explores the environment, it will exceed the maximum navigation steps limitation and then be forced to stop, hindering success in intricate exploration scenarios. It's important to note that previous methods also frequently struggle with these challenging cases. In our future work, we plan to enhance SID's capabilities of error correction and efficient exploration to further boost success rate in difficult scenarios.

Moreover, we acknowledge the following limitations and directions for future work. Firstly, there is potential embodiment gap as the agent is pretrained on robotic navigation assumptions, which may not directly translate to real-world robot deployment. Further challenges include the discrete-to-continuous environment gap and inherent sim-to-real gap, necessitating additional adaptation for practical robotic applications. Furthermore, SID's successful transfer to VLN tasks hinges on textual descriptions generated by VLMs, which may produce incomplete or inaccurate captions due to hallucinations. And evaluating caption quality in the navigation domain is particularly challenging as it requires assessing the alignment between vision, language, and the actions an agent might take. Developing mechanisms for further verifying and refining these generated captions in the embodied tasks could enhance robustness and fully combine the vision-language alignment capabilities of VLMs to bootstrap unified goal-oriented navigation.

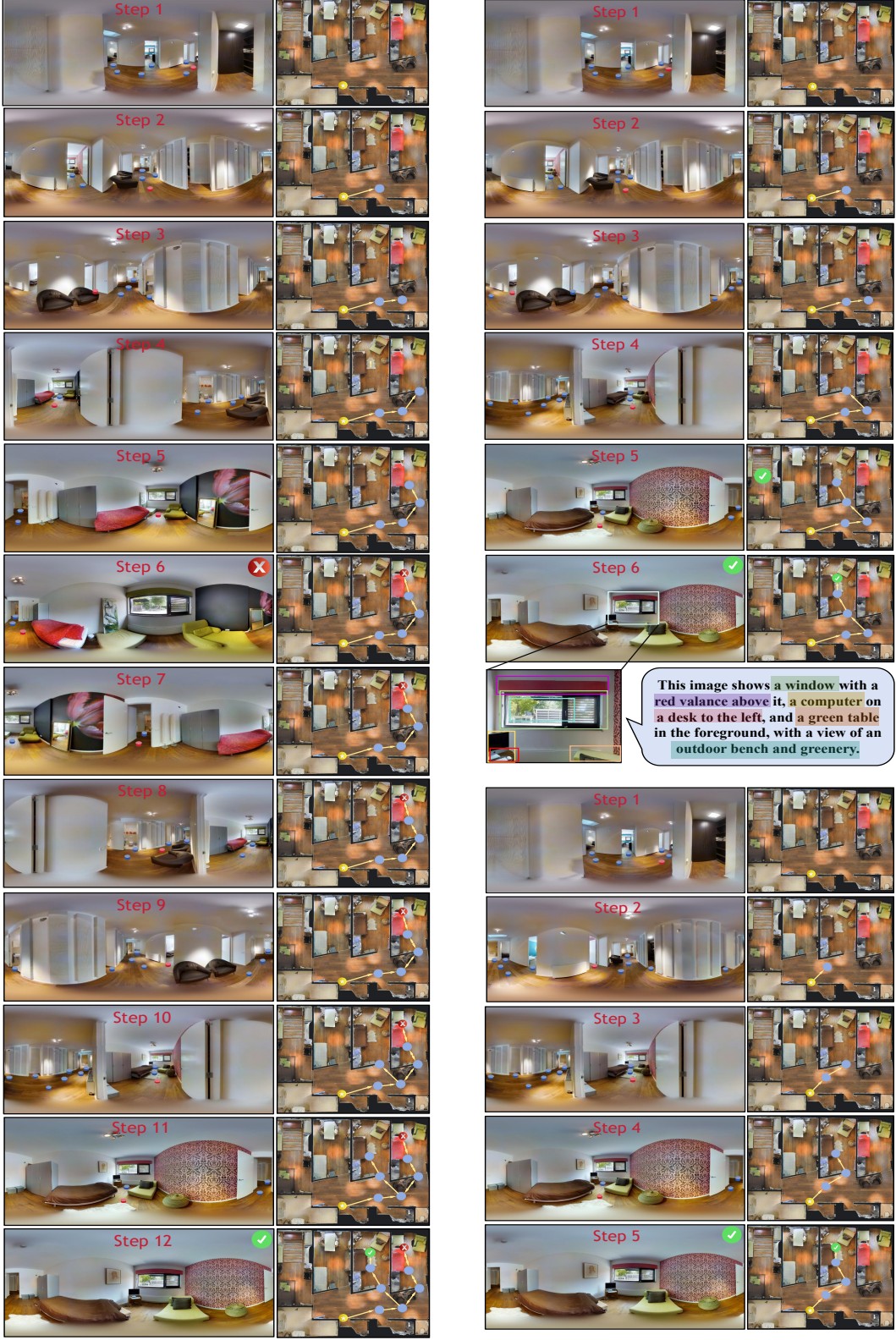

Figure 7: Visualization of the navigation agent's exploration trajectory and the predicted actions at each step. Right-Bottom: The initial shortest-path trajectory towards the navigation target. Left: The navigation agent's exploration and error-correction trajectory across different rooms. Right-Up: The navigation agent's final trajectory and the corresponding image and caption of the target.

