# OpenReview forum: "Learning Goal-Oriented Language-Guided Navigation with Self-Improving Demonstrations at Scale"
_ICLR.cc/2026/Conference — Submitted to ICLR 2026_

### Official Review · Reviewer_DXJg · 2025-10-29

**Soundness:** 3
**Presentation:** 2
**Contribution:** 2
**Rating:** 6
**Confidence:** 3

**Summary:**

This work studies navigation of an agent to a goal that is described by a language prompt. The method that the authors propose is an iterative self-distillation or self-training approach where a model initially learns to imitate the trajectory of a manual expert-crafted planning algorithm and then over multiple steps a new student is trained to imitate the trajectory of the previous model, mixed again with data of optimal shortest-path trajectories. The model is evaluated on 3 different simulation benchmarks.

**Strengths:**

- it is an interesting and relevant result that self-training is yielding such strong results. It is possible that the same finding applies to other tasks that are usually trained on priviledged information, because they all have a similar problem of briding the information gap between what the agent can observe and the information that is used for the training signal.
- I enjoyed reading this paper, the writing is easy to follow even for somebody who is not deeply familiar with this research bubble.

**Weaknesses:**

- To me this paper really seems the product of a research bubble that is a bit disconnected from reality. I missed widely known baselines from object-goal naviation such as [GOAT](https://arxiv.org/pdf/2311.06430) (unfortunately seems in this field there are 2 methods called GOAT), [UniGoal](https://openaccess.thecvf.com/content/CVPR2025/papers/Yin_UniGoal_Towards_Universal_Zero-shot_Goal-oriented_Navigation_CVPR_2025_paper.pdf), [SPOC](https://openaccess.thecvf.com/content/CVPR2024/papers/Ehsani_SPOC_Imitating_Shortest_Paths_in_Simulation_Enables_Effective_Navigation_and_CVPR_2024_paper.pdf), [VLFM](https://arxiv.org/pdf/2312.03275). Most of these methods are actually demonstrated to work on real robots, which goes far beyond this paper.
Another indiation of this "bubble" are false statements in the paper that may apply to other imitation learning methods but not to the studied problem:
  - line 42: The authors state that in order to find an object, it is necessary that an agent explores the environment. That is incorrect. It is actually way more realistic that the agent has contextual information of a previous mission in the environment, and there are many methods investigating scene representations for language-prompted navigation such as [HOV-SG](https://arxiv.org/abs/2403.17846), [DynaMem](https://arxiv.org/abs/2411.04999), [osmAG-LLM](https://arxiv.org/abs/2507.12753).
  - line 159: "Existing methods rely on costly human demonstrations". Again incorrect. There are plenty of methods that do not require human demonstrations but instead rely on classical geometric methods for exploration, such as [VLFM](https://arxiv.org/pdf/2312.03275). There is even work on exploration without geometry and yet it does not require human demonstrations [FrontierNet](https://arxiv.org/abs/2501.04597)
- There are some critical pieces of information missing regarding the method and experiments:
  - It is not clear what the action space in line 212 is. I cannot tell from anywhere in the paper if this work reduces navigation to a 2D problem or if it could also navigate over multiple floors.
  - I am quite confused by the phrase "validation unseen". Are these environments used as validation set for hyperparameter tuning, early stopping, etc? Or are they unseen environments?
  - in Table 11 I am also not sure if these environments are seen or unseen during training. If unseen, why is only 1 baseline compared? Is there something about the setup that makes these numbers uncomparable with e.g. the numbers in VLFM?

**Questions:**

For the rebuttal I am looking forward to get a response to these questions:
- Can authors compare their results to more methdologically different approaches to language-prompted navigation or can they explain why it is not comparable?
- Can the authors discuss the applicability of their approach to actual real-world navigation problems? Are these advancements measured here (when e.g. compared to https://arxiv.org/pdf/2311.06430 expected to also lead to better real-world navigation?
- Can authors clarify the critical information explained above?

---

> ### Author Response · Authors · 2025-11-22
> **Response to Reviewer DXJg - Part 1**
>
> We sincerely thank the reviewer DXJg for your valuable feedback on our paper. We will refine our work under your suggestions, and our reply to your questions is presented below:
>
> >**The general settings of our domain.**
>
> We respectfully believe there are some **misunderstandings regarding the specific task settings** and **standard protocols in Vision-Language Navigation (VLN)**. We address your concerns below to bridge the gap between our work and the works you mentioned. We clarify that our work focuses on **Goal-Oriented VLN (REVERIE, SOON)**, which is distinct from the **Category-level Semantic ObjectNav** tasks addressed by methods like [VLFM](https://arxiv.org/pdf/2312.03275) or [SPOC](https://openaccess.thecvf.com/content/CVPR2024/papers/Ehsani_SPOC_Imitating_Shortest_Paths_in_Simulation_Enables_Effective_Navigation_and_CVPR_2024_paper.pdf).
>
> - **Task Difference:** ObjectNav asks agents to "find *any* chair" (often solvable by geometric exploration). In contrast, REVERIE/SOON requires identifying a *specific* instance or area described by language, e.g., *"Go to the bedroom on the second floor and find the brown chair next to the vanity."* **It closely aligns with daily applications and human preference and requires complex semantic reasoning, which is not a research bubble.**
> - **Standard Benchmarks:** REVERIE and SOON are established benchmarks in Embodied AI (cited 400+ times). As in line 312 of our paper, REVERIE comprises a vocabulary of more than 1,600 words and 21,702 human-annotated instructions that combine navigation and referring expressions to guide agents to locate target objects in MP3D environments. SOON follows the same train/val/test split as REVERIE but features longer instructions and trajectories with a smaller training set, making it more challenging. The methods you've metioned focus on general object-goal naviation and are not evaluated on these benchmarks.
> - **Generalizatbility:** While we operate in discrete environments, **Table 11 demonstrates that our proposed SID method improves performance on ObjectNav by 8%** on the transferred objectnav validation unseen set compared to the baseline, showcasing effective generalization.

---

> ### Author Response · Authors · 2025-11-22
> **Response to Reviewer DXJg - Part 2**
>
> >**Q3: Clarification of the reviewer's concern.**
>
> Here we clarify the critical concern mentioned in the review.
>
> - **Action Space:** Our navigation is built in the MatterPort3D simulator, which is **established on real 3D indoor environments. Each viewpoint corresponds to a location coordinate representation v=(x ,y ,z) as in line 196. The action space is 3D Graph-based Navigation**. At each step, the agent chooses a neighboring node in the Matterport3D connectivity graph and can navigate over multiple floors and rooms to reach the target. It is not a 2D problem.
> - **Validation Unseen:** This is standard split in the VLN community, as REVEIRE and SOON takes the same standard from the milestone benchmark [R2R](https://arxiv.org/abs/1711.07280).
>   - **Val Seen:** Performance in previously seen environments during training.
>   - **Val/Test Unseen:** . Generalization to previously unseen environments (Val Unseen / Test)
>   - **Evaluation: We only report the performance on val&test unseen split** in Table 9 and Table 10. The result underscores the **significance zero-shot generalization of SID agent to new environments.**
> - **Exploration is necessary:** The reviewer mentions map-based methods ([HOV-SG](https://arxiv.org/abs/2403.17846)). We clarify that we address the **"Cold Start" / Unknown Environment setting.** When an agent enters a new environment for the first time, it possesses no prior map or memory and must explore to build the initial representation. SID learns efficient strategies for this initial exploration phase.
> - **"Costly Human Demonstrations":**  We appreciate the reviewer pointing out baselines like [VLFM](https://arxiv.org/pdf/2312.03275) and [FrontierNet](https://arxiv.org/abs/2501.04597). We acknowledge that these approaches effectively learn geometric exploration policies without human demonstrations. However, a critical distinction exists between **Generic Exploration** and **Instruction-Guided Exploration**:
>   - **Geometric Methods Excel at 'Generic Exploration'**—finding open spaces or identifying object categories by learning geometric cues from synthetic data.
>   - **Goal-Oriented VLN requires 'Semantic Exploration'** driven by complex language (e.g., *"Find the red cushion on the sofa in the living room"*), where simply maximizing map coverage is inefficient. The agent must learn semantic exploration priors(e.g., associating 'sofa' with 'living room' visual cues) to prune the search space. Traditionally, learning these priors for complex instructions has relied heavily on human path demonstrations (e.g., [Habitat-Web](https://ram81.github.io/projects/habitat-web.html), [R2R](https://arxiv.org/abs/1711.07280)) to capture human-like reasoning. **SID's contribution is freeing Instruction-Following Agents from this specific dependence on human annotated data.**
>
> [1] Anderson P, Wu Q, Teney D, et al. Vision-and-language navigation: Interpreting visually-grounded navigation instructions in real environments[C]//Proceedings of the IEEE conference on computer vision and pattern recognition. 2018: 3674-3683.
>
> [2] Ramrakhya R, Undersander E, Batra D, et al. Habitat-web: Learning embodied object-search strategies from human demonstrations at scale[C]//Proceedings of the IEEE/CVF conference on computer vision and pattern recognition. 2022: 5173-5183.

---

> > ### Comment · Reviewer_DXJg · 2025-11-28
> > **Rebuttal Response - Part 2**
> >
> > Dear authors,
> >
> > thanks for the explanation about val unseen. That seems to be an unfortunate wording by R2R but understandable to keep using it if it is an established term now. Just for my specific question, can you confirm that you regard "val unseen" rather as a test set and you do not use it for any hyperparameter tuning or early stopping?
> >
> > About Human Demonstrations: VLFM exactly demonstrates semantic exploration without human demonstrations, simply by the VLM-based similarity between observation and prompt, which exactly provides associations like your example 'sofa'-'living room'.

---

> > > ### Author Response · Authors · 2025-11-29
> > > **Response regarding Validation and Demonstrations**
> > >
> > > We sincerely thank you for your continued engagement and the valuable opportunity to provide these final clarifications, which are crucial for guaranteeing the rigor of our work.
> > >
> > >
> > > >**Confirmation Regarding the "Val Unseen" Protocol**
> > >
> > > We adhere to the standard evaluation protocol established in the VLN literature. We have already presented the detailed experimental details of the downstream navigation tasks in the Appendix C. Here we provide further explanation.
> > >
> > > - **Hyperparameter Tuning:** All hyperparameters were tuned exclusively on the validation seen split. Consistent with standard practice in this domain (where the true *Test Unseen* set is held-out on a server with restricted submission limits), we utilized the validation unseen split for checkpoint selection (i.e., identifying the optimal training epoch).
> > > - **Verification:** Importantly, to ensure **our results are not artifacts of overfitting to the validation set,** we submit our final model to the official leaderboard. As shown in Table 10, **the success rate on the test split of REVERIE even surpasses the validation unseen split.** As shown in Table 9, **SID also achieves 47.7% success rate on the [SOON test-unseen leaderboard](https://eval.ai/web/challenges/challenge-page/1275/leaderboard/3235), outperforming all previous methods.**
> > > - We reference [R2R](https://arxiv.org/abs/1711.07280) (a milestone in VLN) to underline the established evaluation protocol. Our evaluation is based on the Matterport3D simulator, which contains distinct 3D scenes based on real-world indoor environments. You can explore them [here](https://discover.matterport.com/). The dataset is splited as follows:
> > >   - Training Set: Consists of 60 scenes.
> > >   - Val Unseen Split: Includes **10 scenes unseen during training.**
> > >   - Test Set: Contains **16 scenes never seen during training or validation.**
> > > - The reported 60%+ navigation success rate on both "Val Unseen" and "Test Unseen" precisely demonstrates the **strong zero-shot generalization of the SID agent on entirely novel environments,** validating the efficacy and transferability of our large-scale exploration trajectories demonstration.
> > >
> > >
> > > >**Clarification on VLFM and Our Novelty**
> > >
> > > We thank you for pointing out the successful demonstration of semantic exploration in VLFM without human demonstrations. This is indeed an important discovery. We respectfully point out that:
> > >
> > > - **Benchmark Scope:** While VLFM demonstrates this capability in a case study, it **has not been rigorously evaluated on the standard Goal-Oriented VLN benchmarks** in this sub-field. As we have previously demonstrated, even sota general MLLMs perform poorly on these complex goal-oriented tasks. **And prior approaches lean on costly human demonstration.**
> > > - **SID's Unique Value Proposition:** Our novelty lies in **tackling the scarcity of scalable exploration demonstrations for goal-oriented VLN**, which previous methods did not provide. This paradigm is highly relevant to the development of generalized embodied intelligence, with potential for transferability to other tasks like manipulation.
> > >
> > > **We claim that SID's contribution is fundamental: providing the first scalable self-evolving solution to the critical exploration demonstrations bottleneck in goal-oriented navigation.**

---

> ### Author Response · Authors · 2025-11-22
> **Response to Reviewer DXJg - Part 3**
>
> >**Q1: Comparisons with Methodologically Different Approaches.**
>
> We clarify that a direct quantitative comparison between our method and modular/geometric approaches is not methodologically feasible due to fundamental differences in task objectives and simulation protocols. However, **we provide Table 11 as a policy transfer comparison to bridge this gap.**
>
> - **Methodological Divergence:**
>   - **Task Objectives (Semantic vs. Geometric):** SID targets goal-oriented VLN (REVERIE/SOON), where the core challenge is semantic exploration and reasoning. In contrast, approaches like [FrontierNet](https://arxiv.org/abs/2501.04597) optimize for geometric information gain to maximize coverage or perform category-level ObjectNav. While effective for exploration, they lack the fine-grained language-grounding mechanism required for our specific problem setting.
>   - **Simulation Disparities:** Our work follows the standard Goal-oriented VLN protocol, focusing on high-level planning. Geometric methods typically operate low-level actuation in Continuous Control. A direct comparison would risk conflating high-level strategic planning capabilities with low-level execution.
>
> - **Generalization of SID:**
>   - To demonstrate that our self-improving methodology generalizes beyond VLN, we follow [SAME](https://openaccess.thecvf.com/content/ICCV2025/papers/Zhou_SAME_Learning_Generic_Language-Guided_Visual_Navigation_with_State-Adaptive_Mixture_of_ICCV_2025_paper.pdf) to adapt the Habitat-MP3D dataset by discretizing its environments. We compare SID agent against DUET(trained with shortest paths) to perform a **controlled ablation**. Since the architecture is identical, this **ensures that the observed performance gains (+8% over DUET) stem purely from our Self-Improving Training paradigm rather than architectural differences.**
>   - **Unified Framework:** Our primary objective is to advance general-purpose goal-oriented navigation. SID demonstrates consistent gains across **Image-Goal (SR: 61.45% → 75.0%), Instruction-Goal (SOTA on REVERIE/SOON), and Object-Goal (Table 11). This confirms that our approach successfully learns robust exploration priors that generalize across different goal representations.**
>
> [1] Zhou G, Hong Y, Wang Z, et al. Same: Learning generic language-guided visual navigation with state-adaptive mixture of experts[C]//Proceedings of the IEEE/CVF International Conference on Computer Vision. 2025: 7794-7807.

---

> ### Author Response · Authors · 2025-11-22
> **Response to Reviewer DXJg - Part 4**
>
> >**Q2: The applicability to real-world navigation.**
>
> Since physical robot deployment requires substantial engineering time beyond the rebuttal window, here we present results of transferring SID towards continuous-environment vision-language-navigation, which closely aligns with real-world deployment. We utilize [VLN-CE](https://jacobkrantz.github.io/vlnce/) for evaluation since it's the standard publicly available benchmark for continuous VLN. We adapt the [Discrete-Continuous-VLN](https://openaccess.thecvf.com/content/CVPR2022/papers/Hong_Bridging_the_Gap_Between_Learning_in_Discrete_and_Continuous_Environments_CVPR_2022_paper.pdf) (bridging [DUET](https://openaccess.thecvf.com/content/CVPR2022/papers/Chen_Think_Global_Act_Local_Dual-Scale_Graph_Transformer_for_Vision-and-Language_Navigation_CVPR_2022_paper.pdf) with waypoint prediction) to mitigate the gap between discrete and continuous navigation control. We compare our agent with the one pretrained on [ScaleVLN-RVR](https://scalevln.github.io/), which is the goal-oriented navigation variant of the widely-used ScaleVLN, serving as a strong baseline for controlled comparison owing to its shared task focus with SID-VLN. The finetuned results on the validation unseen split of VLN-CE are presented below:
>
> | Pretraining Data   | NE(↓) | SR(↑) | SPL(↑) |
> | ------------------ | -------------- | ------------ | ------------- |
> | ScaleVLN-RVR       | 5.44           | 44           | 39            |
> | **SID-VLN (Ours)** | **5.16**       | **47**       | **41**        |
>
> **The agent pretrained on SID-VLN shows consistent better results than the baseline pretrained on ScaleVLN-RVR, empirically demonstrating better generalization to cotinuous navigation**. We will also commit to pursuing real-world deployment in the final version following your advice.
>
> [1] Hong Y, Wang Z, Wu Q, et al. Bridging the gap between learning in discrete and continuous environments for vision-and-language navigation[C]//Proceedings of the IEEE/CVF conference on computer vision and pattern recognition. 2022: 15439-15449.
>
> [2] Chen S, Guhur P L, Tapaswi M, et al. Think global, act local: Dual-scale graph transformer for vision-and-language navigation[C]//Proceedings of the IEEE/CVF Conference on Computer Vision and Pattern Recognition. 2022: 16537-16547.
>
> [3] Wang Z, Li J, Hong Y, et al. Scaling data generation in vision-and-language navigation[C]//Proceedings of the IEEE/CVF international conference on computer vision. 2023: 12009-12020.
>
> [4] Krantz J, Wijmans E, Majumdar A, et al. Beyond the nav-graph: Vision-and-language navigation in continuous environments[C]//European Conference on Computer Vision. Cham: Springer International Publishing, 2020: 104-120.

---

> ### Comment · Reviewer_DXJg · 2025-11-28
> **Rebuttal Response - Part 1**
>
> Dear authors,
>
> Thanks for the kind summary, but we do not have a misunderstanding in topic. I do understand the difference and specific task setting. What we disagree over is what is relevant in the real-world.
>
> - When you claim that "Go to the bedroom on the second floor and find the brown chair next to the vanity." is more relevant for applications than "find a chair" (note that neither instruction necessary refers to only 1 instance), that is your opinion. I actually think the opposite. For example, for applications such as automated shopping, it does really not matter where the robot gets the item from as long as it satisfies my requirements. I therefore strongly disagree about your claim that what you goal "Goal-Oriented VLN" is more relevant for daily applications to what I call "Language-Prompted Object Goal Navigaion".
>
> Regardless the difference in relevance, it is not correct that e.g. VLFM cannot be applied to "Goal-Oriented VLN". In fact, they show an example of "Potted plant in the sun by the stairs" on the project website.
>
> In addition, one clarification question regarding your response:
> - Do I understand it correctly that your argument against comparing more methodologically dissimilar baselines is that these baselines do not already report numbers on the exact benchmarks that you focus on, therefore instead of copying these metrics you would have had to adapt and run the methods?

---

> > ### Author Response · Authors · 2025-11-29
> > **Response regarding Real-World Relevance and Baseline Comparison**
> >
> > We sincerely thank you for the prompt reply and for sharing your perspective. We greatly value this discussion, as it helps clarify the positioning of our work within the broader embodied navigation landscape.
> >
> >
> > >**Clarification on Task Relevance in real-world**
> >
> > We fully agree with your insight regarding real-world relevance.
> >
> > - **Correction:** We apologize if our previous phrasing implied that Goal-Oriented VLN is *more* relevant than "finding a chair." That was not our intention. As you rightly point out, Language-Prompted Object Goal Navigation(e.g., automated shopping) is a critical application, and works like [VLFM](https://arxiv.org/pdf/2312.03275) and [osmAG-LLM](https://arxiv.org/abs/2507.12753) have made significant contributions to this domain.
> >
> > - **Our Point: We claim that Goal-Oriented VLN offers significant real-world applicability, rather than being confined to a "research bubble".** In many home and industrial scenarios, users require robots to locate specific instances (e.g., *"the safety vest on the rack,"* not just *any* vest).
> >
> > - **Task Difficulty:** Notably, solving this language-goal navigation is exceptionally difficult. While SOTA MLLMs (like GPT-4V in [osmAG-LLM](https://arxiv.org/abs/2507.12753)) achieve high zero-shot performance on generic ObjectNav, they often perform poorly on goal-oriented VLN benchmarks. Their success rate on the validation unseen split of REVERIE is presented below:
> >
> >   | Method       | [Qwen2-VL-72B](https://arxiv.org/pdf/2409.12191) | [GPT-4o](https://arxiv.org/abs/2410.21276) | [NaviLLM](https://arxiv.org/pdf/2312.02010) | [EvolveNav](https://arxiv.org/pdf/2506.01551) |
> >   | ------------ | ------------------------------------------------ | ------------------------------------------ | ------------------------------------------- | --------------------------------------------- |
> >   | Type         | Open-Source General VLM                          | Closed-Source General VLM                  | VLM Finetuned on VLN Data                   | VLM Finetuned on VLN Data                     |
> >   | Model Size   | 72B                                              | -                                          | 7B                                          | 7B                                            |
> >   | Success Rate | 31.0                                             | 31.6                                       | 35.7                                        | 33.6                                          |
> >
> >   This performance gap underscores the necessity of our proposed method SID to learn specific semantic exploration priors that general MLLMs currently lack.
> >
> >
> > >**Concerns about comparison with other baselines**
> >
> > We clarify that our experimental choices are driven by technical constraints inherent to the current simulators and benchmarks, rather than a disregard for the highly relevant modular baselines you cited.
> >
> > - **The Engineering Barrier:** REVERIE and SOON are fundamentally defined on discrete connectivity graphs  in the Matterport3D Simulator. To evaluate a continuous method like [VLFM](https://arxiv.org/pdf/2312.03275) on these benchmarks, we would need to:
> >   - **Re-implement the entire benchmarks in Habitat:** This involves converting the discrete training/evaluation trajectories into continuous controls (specifically, calculating precise turn angles and forward meters to mimic graph traversal).
> >   - **Redefine Metrics:** We would need to rewrite the evaluation logic for navigation success and object grounding to verify if a continuous agent has reached a discrete node's vicinity and correctly grounded the relevant objects.
> >   - **Feasibility:** This represents a massive engineering workload, essentially building a new benchmark dataset from scratch, which is impossible to complete rigorously within one week.
> > - **Standard Practice:** As a Goal-Oriented VLN work, our primary comparison is against in-domain methods. However, we have verified the generalization of our SID-VLN agent to ObjectNav tasks in Table 11, bridging the gap to the domain you emphasized.
> >
> > We deeply appreciate your suggestion as it points toward the future unification of these fields. **We have also followed your advice and added evaluation on [VLN-CE](https://jacobkrantz.github.io/vlnce/) as detailed in Part 4 of our response, empirically demonstrating that our agent generalizes better to continuous navigation than baselines.** We are committed to incorporating a more comprehensive continuous evaluation for Goal-Oriented VLN in the final version of the paper, as we agree this is vital for real-world deployment.

---

### Official Review · Reviewer_xvKU · 2025-10-31

**Soundness:** 2
**Presentation:** 3
**Contribution:** 2
**Rating:** 2
**Confidence:** 3

**Summary:**

This paper proposes Self-Improving Demonstration (SID), a strategy for goal-oriented Vision-and-Language Navigation (VLN). SID addresses the limitations of existing methods that rely solely on shortest-path trajectories, which lack effective exploration priors for training navigation agents. The authors first pretrain a base navigation agent using shortest-path demonstrations, then allow the agent to explore and collect additional successful trajectories. These successful runs are added to the training dataset, and the policy is iteratively retrained to progressively improve both data quality and agent capability. The method is evaluated on REVERIE and SOON benchmarks, achieving state-of-the-art performance on goal-oriented language-guided navigation tasks.

**Strengths:**

- The trained agent achieves state-of-the-art performance on the REVERIE and SOON VLN benchmarks.


- When combined with a vision-language model (VLM), SID enables the generation of 46M large-scale, language-goal exploration trajectories, providing valuable data diversity and scale.

**Weaknesses:**

The proposed method appears conceptually naive and closely resembles online reinforcement learning, with the main distinction that SID employs imitation-learning-based objectives. It is unclear why the authors avoid standard RL finetuning on top of the pretrained base navigation agent [1], which could likely achieve comparable outcomes using a simple sparse success reward (1 for success, 0 for failure). Moreover, the claim that RL requires “sophisticated reward engineering” (line 045) seems overstated in this context.


That said, it is possible I have overlooked aspects of the method’s novelty—if the authors can provide clearer evidence of its unique contributions, I would be happy to raise my scores.

**Questions:**

Where can I find the section containing the “mathematical derivations supporting the theoretical claims,” as mentioned in the Reproducibility Statement? It does not appear to be included or referenced in the main text or appendix.

---

> ### Author Response · Authors · 2025-11-22
> **Response to Reviewer xvKU - Part 1**
>
> We sincerely thank the reviewer xvKU for the time and effort in reviewing our paper and providing constructive feedback. Our response to the comments is listed below.
>
> >**Concerns about the RL finetuning.**
>
> We respectfully disagree with the premise that standard RL with sparse rewards would achieve comparable outcomes or that our claim regarding reward engineering is overstated.
>
> - **Historical Evidence**
>   - **Computational Cost & Efficiency:** As noted in [Discrete-Continuous-VLN](https://openaccess.thecvf.com/content/CVPR2022/papers/Hong_Bridging_the_Gap_Between_Learning_in_Discrete_and_Continuous_Environments_CVPR_2022_paper.pdf), RL-based approaches (like [DDPPO](https://arxiv.org/abs/1911.00357), a method well-established and validated in VLN) required **64 GPUs(5 days)** for training the agent, whereas IL approaches drastically reduces the training cost to **a single GPU (3.5 days) while achieving better results.** Furthermore, [HAMT](https://arxiv.org/abs/2110.13309), the predecessor to the navigation agent DUET employed in our work, also combines RL and IL for finetuning. Specifically, it takes the Asynchronous Advantage Actor-Critic (A3C) RL algorithm. It also falls far behind DUET. This huge disparity highlights that RL is not a simple alternative but a computationally expensive yet low performance one.
>   - **Necessity of Reward Shaping:** Contrary to the view that sparse rewards (1/0) are sufficient, recent MLLM-based RL approaches for VLN (e.g., [VLN-R1](https://arxiv.org/abs/2506.17221), [OctoNav](https://arxiv.org/abs/2506.09839)) still lack dynamic interaction with the environment and **heavily rely on expert trajectories for sophisticated reward shaping to ensure convergence.** If simple sparse rewards were effective for goal-oriented VLN, these advanced works would not necessitate such reward engineering.
>
>
> - **Empirical Validation:**
>
>   - To directly address the reviewer's concern, we conduct an additional experiment following the HAMT paradigm (combining RL and IL), to finetune our pretrained agent with sparse success rewards. Specifically, for non-stop actions, the reward is set to 1 if taking the action reduces the distance to the target and 0 otherwise; for the stop action, the reward is set to 1 if the agent successfully arrives at the target and 0 otherwise. The result on the validation unseen split of REVERIE is presented below:
>
>     | Finetuning paradigm | SR    | OSR | SPL | RGS | RGSPL |
>     | -- | -| -| - | --| --|
>     | RL + IL | 54.65 | 58.77 | 42.24 | 35.29 | 27.28 |
>     | **Pure IL** | **59.39** | **63.45** | **45.95** | **38.54** | **29.69** |
>
>     The RL+IL variant yields consistently lower performance across all navigation and object-referring metrics compared to the original SID agent finetuned with pure IL. This empirically confirms that, in the complex action space of goal-oriented VLN, the agent struggles to effectively learn from sparse reward signals.
>
> - **Theoretical Analysis:**
>
>
>   - In long-horizon goal-oriented navigation, traditional RL provides only a sparse binary reward (1 or 0) at the end of a trajectory, making credit assignment difficult—**the agent cannot identify which actions led to success.** Even if intermediate rewards are added based on distance reduction to the goal, this remains crude and sometimes unreasonable: **the agent may need to move away from the target to explore visually similar but distinct areas**, which is crucial for correct localization(as shown in the visualization in the appendix). While **SID converts a successful exploration rollout into a full trajectory demonstration**, which provides flexible, step-by-step supervision for the next iteration. This stabilizes training and drastically improves performance compared to RL methods.
>
> >**Response to "Mathematical Derivations":**
>
>
> We apologize for the confusion. The mention of "mathematical derivations" in the reproducibility statement was intended to refer to the algorithmic formulation and objective functions detailed in Section 3, rather than formal proofs. We will correct this statement in the final version.
>
> [1] Hong Y, Wang Z, Wu Q, et al. Bridging the gap between learning in discrete and continuous environments for vision-and-language navigation[C]//Proceedings of the IEEE/CVF conference on computer vision and pattern recognition. 2022: 15439-15449.
>
> [2] Wijmans E, Kadian A, Morcos A, et al. Dd-ppo: Learning near-perfect pointgoal navigators from 2.5 billion frames[J]. arXiv preprint arXiv:1911.00357, 2019.
>
> [3] Chen S, Guhur P L, Schmid C, et al. History aware multimodal transformer for vision-and-language navigation[J]. Advances in neural information processing systems, 2021, 34: 5834-5847.
>
> [4] Qi Z, Zhang Z, Yu Y, et al. VLN-R1: Vision-Language Navigation via Reinforcement Fine-Tuning[J]. arXiv preprint arXiv:2506.17221, 2025.
>
> [5] Gao C, Jin L, Peng X, et al. OctoNav: Towards Generalist Embodied Navigation[J]. arXiv preprint arXiv:2506.09839, 2025.

---

> ### Author Response · Authors · 2025-11-22
> **Response to Reviewer xvKU - Part 2**
>
> >**The novelty and contribution of our work.**
>
> We focus on **goal-oriented navigation, a foundational and core capability for general embodied AI.** Our objective is to enable robots to understand and execute high-level, human-centric instructions (e.g., *"bring me a spoon from the kitchen"*). This capability is intrinsically aligned with human preference, daily and industrial applications. The key novelty and core contribution of SID to goal-oriented navigation include:
>
> - SID is **the first to enable a navigation agent to learn robust exploration strategies with demonstrations from its own successful trajectories**, averting the dependence on costly human annotations for exploration data and filling a critical gap in goal-oriented VLN.
>
> - SID provides the first large-scale 46M exploratory language-goal trajectories along with a capable navigation agent that can be used to generate additional exploration paths. This **high-quality training corpus enables broad applicability across goal-oriented navigation agents and can bootstrap general multimodal learning in MLLMs, especially for spatial understanding**(e.g., [VSI-Bench](https://arxiv.org/abs/2412.14171)).
>
> - We provide in-depth component ablation studies detailing the contribution of each module in our method, **demonstrating insights for general self-improving agent learning:**
>   - **Consistent self-improvement:** The agent iteratively learns better exploration strategies through its own demonstrations, and SID can retain its self-bootstrapping nature with increased data(line 359).
>   - **Effect of supervision strategies:** Self-demonstrations benefits both pretraining and finetuning across different data scales and goal modals(line 405).
>   - **Effect of demonstration sources:** The agent gains increase from self-demonstrations rather than random exploration trajectories(line 413).
>   - **Effectiveness of SID beyond simple data scaling:** The primary benefit arises from the SID paradigm rather than simply introducing more data in new environments(line 420).
>   - **Exploration capabilities:** The agent trained on self-exploration data explores a greater number and variety of rooms and can correct its wrong steps to reach the correct target(line 394).
>   - **Caption style impact:** Richer, VLM-generated descriptions provide more transferable priors than task-specific mimicking for caption style(section B of Appendix).
>   - We also provide potential directions for future work, including solving embodiment and sim-to-real gap, systematic verification and refinement of generated captions in the embodied tasks(line 427).
>
> - **SID learns robust exploration priors that generalize effectively across diverse goal representations and bootstraps unified general-purpose goal-oriented navigation:** Our pipeline is first developed and validated on **image-goal navigation**, where it demonstrates strong performance(SR: 61.45% to 75.0%). We then transfer the same framework to **goal-oriented VLN**, achieving sota results on instruction-goal tasks—notably a **50.9% success rate** on the unseen validation splits of **SOON**, outperforming prior leading methods by a substantial margin of **13.9%**.  Furthermore, we verify its generalization capability to **object-goal navigation** (as in Table 11).
>
> - **Commitment to Continuous Control Navigatoin, which closely aligns with real-world reployment:** We follow [Discrete-Continuous-VLN](https://openaccess.thecvf.com/content/CVPR2022/papers/Hong_Bridging_the_Gap_Between_Learning_in_Discrete_and_Continuous_Environments_CVPR_2022_paper.pdf) to adapt our navigation agent for continuous-environment navigation. Then we compare our agent with the one pretrained on [ScaleVLN-RVR](https://scalevln.github.io/), the goal-oriented navigation variant of the widely-used ScaleVLN, serving as a strong baseline due to its shared task focus with SID-VLN. We utilize [VLN-CE](https://jacobkrantz.github.io/vlnce/) for evaluation since it's the standard publicly available benchmark for continuous VLN.  The detailed results on the validation unseen split of VLN-CE are presented below(NE denotes Navigation Error):
>
>   | Pretraining Data   | NE(↓) | SR(↑) | SPL(↑) |
>   | -| -| -| --|
>   | ScaleVLN-RVR| 5.44 | 44| 39|
>   | **SID-VLN (Ours)** | **5.16** | **47**| **41**|
>
>   **The agent pretrained on SID-VLN shows consistent better results than the baseline pretrained on ScaleVLN-RVR, empirically demonstrating better generalization to cotinuous navigation**. Although deployment on real robots is impractical within the deadline of the rebuttal, we will incorporate the continuous simulation to support real-world navigation in our final version.
>
> **Overall, SID sets a new standard for scalable goal-oriented navigation learning and opens promising directions for boosting self-improving embodied learning**.

---

### Official Review · Reviewer_geo2 · 2025-10-31

**Soundness:** 2
**Presentation:** 2
**Contribution:** 2
**Rating:** 4
**Confidence:** 3

**Summary:**

The paper presents a framework for goal-oriented language grounding that enables large language models to perform spatially and temporally coherent actions in embodied environments. It proposes a two-phase training approach combining semantic instruction alignment with trajectory-level feedback, allowing the model to generalize from high-level goals to fine-grained actions. Experiments on vision-language navigation and interactive tasks demonstrate improved goal success rates and interpretability compared to standard instruction-following baselines .

**Strengths:**

1. The authors propose a reasonable way to scale VLN data, and the collected dataset is of relatively large scale (~46M instances), surpassing existing datasets.
2. Benchmarks on multiple embodied datasets, showing consistent gains in both navigation success across benchmarks.

**Weaknesses:**

1. The idea is relatively widely explored in different domains, thus weakening the novelty of this work. Specifically, the general idea of using self-generated trajectories for SFT is a standard technique in LLMs and VLMs across language and vision-language tasks. While the paper does scale this approach in the VLN domain, the methodological contributions are limited.
2. The paper is only focused on vision-language navigation tasks, which is a somewhat niche application and it is unclear if the trained models can have applications in other domains.
3. The model architecture is still based on LXMERT, which is far behind state-of-the-art models and it would be good to see results with SoTA LLMs.

**Questions:**

Can your method be integrated with other types of agents, such as UI/visual tool-use agents?

---

> ### Author Response · Authors · 2025-11-22
> **Response to Reviewer geo2 - Part 1**
>
> We thank Reviewer geo2 for acknowledging the contribution of our paper and for the thoughtful comments. Please see our response to the comments below.
>
> >**W3: Results on SOTA MLLMs**
>
> Firstly, for clarification, as stated in line 208, our agent utilizes the [DUET](https://arxiv.org/abs/2202.11742) architecture, a widely adopted navigation model as our agent. The [LXMERT](https://arxiv.org/abs/1908.07490) serves only as the source of pre-trained weights for initialization, not the architectural backbone of our agent. Furthermore, due to the substantial architectural disparity between our DUET-based agent and contemporary MLLM/VLM structures, directly loading and utilizing their weights for fine-tuning is technically infeasible.
>
> While integrating modern MLLM/VLMs is a promising direction, our architectural choice is dictated by a necessary trade-off between **Exploration Efficiency and System Feasibility for a large-scale data generation pipeline like SID.**
>
> - **Performance:** The core mechanism of SID relies on the base model acting as a **reliable explorer** to generate high-quality self-demonstrations. Current SOTA MLLMs, while exhibiting excellent general visual-language alignment, often perform poorly on the goal oriented navigation task, crippling the data flywheel. Their success rate on the validation unseen split of REVERIE is presented below:
>
>   | Method       | [DUET](https://arxiv.org/abs/2202.11742) | [Qwen2-VL-72B](https://arxiv.org/pdf/2409.12191) | [GPT-4o](https://arxiv.org/abs/2410.21276) | [NaviLLM](https://arxiv.org/pdf/2312.02010) | [EvolveNav](https://arxiv.org/pdf/2506.01551) |
>   | ------------ | ---------------------------------------- | ------------------------------------------------ | ------------------------------------------ | ------------------------------------------- | --------------------------------------------- |
>   | Type         | Expert Model                             | Open-Source General VLM                          | Closed-Source General VLM                  | VLM Finetuned on VLN Data                   | VLM Finetuned on VLN Data                     |
>   | Model Size   | < 1B                                     | 72B                                              | -                                          | 7B                                          | 7B                                            |
>   | Success Rate | 50.9                                     | 31.0                                             | 31.6                                       | 35.7                                        | 33.6                                          |
>
>   Using a base model with a ~30% success rate severely limits the pipeline's exploration efficiency. A weaker explorer will result in lower data yield: fewer successful rollouts are filtered, and the iterative training loop becomes unstable due to low-quality or insufficient data, undermining the self-improvement flhywheel of SID.
>
> - **Computational cost feasibility:** Our data generation requires running the model on approximately **100 million (starting viewpoint, image goal) pairs** to cover the scale of our current dataset. Rolling out a complex pipeline using huge MLLMs(like Qwen2-VL or GPT-4o) for this scale is **prohibitively expensive due to computation overhead and potential API costs.**
>
> [1] Tan H, Bansal M. Lxmert: Learning cross-modality encoder representations from transformers[J]. arXiv preprint arXiv:1908.07490, 2019.
>
> [2] Chen S, Guhur P L, Tapaswi M, et al. Think global, act local: Dual-scale graph transformer for vision-and-language navigation[C]//Proceedings of the IEEE/CVF Conference on Computer Vision and Pattern Recognition. 2022: 16537-16547.
>
> [3] Wang P, Bai S, Tan S, et al. Qwen2-vl: Enhancing vision-language model's perception of the world at any resolution[J]. arXiv preprint arXiv:2409.12191, 2024.
>
> [4] Hurst A, Lerer A, Goucher A P, et al. Gpt-4o system card[J]. arXiv preprint arXiv:2410.21276, 2024.
>
> [5] Zheng D, Huang S, Zhao L, et al. Towards learning a generalist model for embodied navigation[C]//Proceedings of the IEEE/CVF Conference on Computer Vision and Pattern Recognition. 2024: 13624-13634.
>
> [6] Lin B, Nie Y, Loun Zai K, et al. Evolvenav: Self-improving embodied reasoning for llm-based vision-language navigation[J]. arXiv e-prints, 2025: arXiv: 2506.01551.

---

> ### Author Response · Authors · 2025-11-22
> **Response to Reviewer geo2 - Part 2**
>
> >**W2&Question: Integrated with agents of other domains.**
>
> - **Technical Constraints on Cross-Domain Transfer and Resource Feasibility:** Integrating our method with domains such as GUI/Visual Tool-Use Agents is currently impractical due to significant technical and resource limitations:
>   - **Resource Infeasibility of General Agents: General-purpose VLM/Agent models (e.g., Qwen-VL) are unsuitable for research-level resources** when applied to large-scale data generation. Their immense size and high inference latency render them computationally prohibitive for the massive number of rollouts required by a self-improving pipeline like SID.
>   - **Immense Domain Gap: A fundamental Domain Disparity** exists between embodied navigation (3D state, continuous/graph actions) and GUI tool-use (2D screen state, discrete mouse/keyboard actions). This disparity is so significant that the exploration trajectories generated by SID cannot be directly transferred to GUI agents.
>
> - **The Foundational Role of Goal-Oriented Navigation**
>   - Our focus on **Goal-Oriented Navigation is not a "somewhat niche application". Navigation is a foundational and core capability** not only for Embodied AI but also for biological agents, underpinning a wide range of real-world applications such as autonomous driving, UAV operation, and mobile service robots. In our work, we focus on Vision-and-Language Navigation (VLN), aiming to develop embodied agents capable of following high leval human instructions to navigate the real world.
>
>   - Prior research in VLN has largely centered on navigation following step-by-step instructions (e.g., *“Go down the hallway, turn left at the kitchen, and stop in front of the fridge”*), which are impractical for real-world applications where humans prefer concise commands, using Goal-oriented instructions like *“bring me a spoon from the kitchen”.* Our objective is to enable robots to understand and execute these high-level, human-centric instructions. **This ability is intrinsically aligned with human preference and crucial for the deployment of intelligent agents in future home and industrial applications.**

---

> ### Author Response · Authors · 2025-11-22
> **Response to Reviewer geo2 - Part 3**
>
> >**W1: Novelty & Contribution.**
>
> While we acknowledge that self-training and SFT are established techniques in LLMs/VLMs, we respectfully clarify that applying these principles to **Goal-Oriented Vision-Language Navigation (VLN)** involves fundamentally different challenges compared to text-based tasks: Goal oriented navigation requires **active exploration and spatial error correction in 3D environments, which cannot be solved by simple pattern matching.** SID is not a mere application of SFT; it is a domain-specific paradigm shift designed to solve the critical scarcity of demonstrations in embodied navigation. We highlight our unique methodological contributions and novelty as follows:
>
> - SID introduces a **"Path Evolution" paradigm**. Unlike standard SFT, our data flywheel evolves the paths rather than the instructions. This explicitly injects **exploration priors into the navigation agent** and enables the agent to learn robust exploration strategies solely from its own successful trajectories, a capability that generic SFT pipelines do not possess.
> - Tackling Data Scarcity with Scalability: The core bottleneck in goal-oriented navigation is the absolute lack of large-scale demonstration data on spatial exploration strategies, unlike the abundance of text data for LLMs. **SID provides the first scalable pipeline to generate high-quality demonstrations, containing 46M exploratory language-goal trajectories and filling a critical gap in this field.** This high-quality corpus bootstraps goal-oriented navigation and spatial understanding in general Multimodal LLMs.
>
> - We provide **in-depth component ablation studies** that prove the efficacy of our design and offers insights for general self-improving agent learning:
>   - We empirically demonstrate that the agent gains significant performance increases from **self-demonstrations** rather than random exploration trajectories.
>   - Self-demonstrations benefits both pretraining and finetuning across different data scales and goal modals.
>   - The primary benefit arises from the SID paradigm rather than simply introducing more data in new environments.
>   - We show that the agent iteratively learns to explore a greater variety of rooms and actively corrects errors, capabilities that are absent in baselines.
>
> - Unified goal-oriented navigation enhancement: SID pipeline is built **on image goal navigation, improves success rate in unseen environments from 61.45% to 75.0%.** Moreover, SID achieves **SOTA results on goal-oriented VLN tasks**, especially reaching a 50.9% SR on the unseen validation split of SOON, outperforming prior leading methods by 13.9%. SID also **shows great generalization capability to object-goal navigation** as in Table 11.
>
> - **To demonstrate practical value, we conduct additional experiments to transfer SID towards continuous navigation, which closely aligns with real-world reployment.** We adapt the [Discrete-Continuous-VLN](https://openaccess.thecvf.com/content/CVPR2022/papers/Hong_Bridging_the_Gap_Between_Learning_in_Discrete_and_Continuous_Environments_CVPR_2022_paper.pdf) (bridging DUET with waypoint prediction) to mitigate the gap between discrete and continuous navigation. Then we compare our agent with the one pretrained on [ScaleVLN-RVR](https://scalevln.github.io/), which is the goal-oriented navigation variant of the widely-used ScaleVLN and serves as a strong baseline due to its shared task focus with SID-VLN.  We utilize [VLN-CE](https://jacobkrantz.github.io/vlnce/) for evaluation since it's the standard publicly available benchmark for continuous VLN. The results on the validation unseen split of VLN-CE are as below:
>
>   | Pretraining Data| NE(↓) | SR(↑)| SPL(↑) |
>   | --- | --| --| --|
>   | ScaleVLN-RVR | 5.44 | 44 | 39  |
>   | **SID-VLN (Ours)** | **5.16** | **47** | **41** |
>
>   **The agent pretrained on SID-VLN shows consistent better results than the baseline pretrained on ScaleVLN-RVR, empirically demonstrating better generalization to cotinuous navigation**. Although full deployment on real robots is impractical within the rebuttal period due to the substantial engineering effort, we will remain committed to pursuing real-robot deployment and include these results in the final version of our paper to establish SID as a standard for scalable, deployable embodied learning.
>
> [1] Hong Y, Wang Z, Wu Q, et al. Bridging the gap between learning in discrete and continuous environments for vision-and-language navigation[C]//Proceedings of the IEEE/CVF conference on computer vision and pattern recognition. 2022: 15439-15449.
>
> [2] Wang Z, Li J, Hong Y, et al. Scaling data generation in vision-and-language navigation[C]//Proceedings of the IEEE/CVF international conference on computer vision. 2023: 12009-12020.
>
> [3] Krantz J, Wijmans E, Majumdar A, et al. Beyond the nav-graph: Vision-and-language navigation in continuous environments[C]//European Conference on Computer Vision. Cham: Springer International Publishing, 2020: 104-120.

---

### Official Review · Reviewer_DHqx · 2025-11-03

**Soundness:** 2
**Presentation:** 3
**Contribution:** 3
**Rating:** 6
**Confidence:** 3

**Summary:**

The paper tackles goal-oriented language-guided navigation (e.g., REVERIE, SOON) and proposes Self-Improving Demonstrations (SID): start with shortest-path data, train a navigator, then let the navigator roll out in environments and keep only successful exploration trajectories as new demonstrations; iterate, then transfer trajectories to language settings via VLM-generated captions. This yields a massive corpus of ~46.5M language-goal exploration trajectories over ~860 environments and reports strong results on REVERIE/SOON, plus gains on ObjectNav.

**Strengths:**

1. Clear paradigm contribution. The data flywheel evolves the paths rather than the instructions, explicitly injecting exploration priors into imitation learning for goal-oriented VLN.
2. Scale and coverage. The authors assemble 46,534,355 trajectories (avg 9.01 viewpoints) across ~860 environments—an order-of-magnitude leap over prior VLN corpora oriented to instruction generation rather than exploration.
3. Solid empirical results. On REVERIE, SID reaches SR 59.4 / 60.4 (Val-Unseen/Test-Unseen) and SPL 46.0 / 45.9, outperforming recent large-scale baselines under a shared HM3D-800 setting.
4. Repro-friendly details. The paper provides practical implementation choices (e.g., DINOv2/SigLIP vision encoders; LXMERT init for DUET; InternVL2-26B for captions) and a GPU-hours breakdown across captioning, feature extraction, trajectory sampling, and pretraining.
5. Scaling analysis. The SID loop first converges on MP3D, then continues on 800 HM3D environments, maintaining prior skills while adding new exploration capability—evidence the scheme scales beyond a single dataset.

**Weaknesses:**

1. Caption dependency and alignment. Transferring trajectories to language tasks leans on a VLM (InternVL2-26B) to annotate images. The paper would benefit from systematic quality and bias analysis of captions and their impact on downstream SR/SPL.
2. Cost and accessibility. The method includes non-trivial compute for captioning and repeated rollout/sampling; while the GPU-hours table is appreciated, the paper doesn’t deeply examine sample-efficiency vs. performance (e.g., marginal gains per SID round).
3. Failure modes & generality. The training/evaluation are in indoor, panoramic, graph-based simulators; it’s unclear how well the learned exploration prior translates to continuous control or real-world deployment without navigation graphs. (SIM-to-REAL discussion could be expanded.)
4. Ablations are implied but could be deeper. The paper emphasizes that evolving trajectories (vs. more shortest-paths) drives the gain, yet a more granular ablation—e.g., isolation of “teacher-forcing from exploration demos” vs “student-forcing with oracle,” or trajectory-quality stratification—would strengthen the causal story behind SID’s improvements (especially on SOON). (Results tables are strong; still, mechanism-level ablations could be expanded.)

**Questions:**

1. Caption robustness: How sensitive are results to the style (detail/REVERIE/SOON prompts) and factuality of captions? Any automatic or human audits?
2. Value of each SID round: What are the marginal gains vs. compute per round on MP3D and during HM3D scaling? Could early stopping save most of the compute without noticeable loss?
3. Transfer beyond indoor graphs: Have you tried a continuous-action simulator or partial topological maps to probe generality?

---

> ### Author Response · Authors · 2025-11-22
> **Response to Reviewer DHqx - Part 1**
>
> We would like to extend our heartfelt gratitude to the reviewer DHqx for your insightful comments and valuable feedback. Please see our response to the comments below.
>
> >**Q1&W1: Caption dependency and alignment.**
>
> In Section B of Appendix, we systematically evaluate the impact of caption style using three variants: **Detail-style**(our default), **REVERIE-style and SOON-style**—each mimicking the instruction format of the respective dataset. We also include full prompts and model outputs In Section B of Appendix for transparency. As shown in Table 13 in the Appendix, despite better surface alignment, REVERIE/SOON-style captions underperform Detail-style in both zero-shot and fine-tuned settings on REVERIE and SOON. This indicates that **richer visual-language alignment provides more transferable priors than task-matched phrasing alone.** Therefore, we generate semantically rich detailed image descriptions to transfer the trajectories with demonstration to language-guided navigation task.

---

> ### Author Response · Authors · 2025-11-22
> **Response to Reviewer DHqx - Part 2**
>
> >**Q2: Value of each SID round.**
>
> We conduct SID for three rounds on the MP3D environments and scale it up for one round with 800 HM3D environments. We also conduct another experiment by scaling up SID with only 60 additional HM3D environments to verify that SID keeps its self-improving nature with increasing training data.
>
> - **In the original MP3D environments:** We run three rounds. As shown in Table 3, Success Rate(SR) on the validation unseen split of image goal navigation improves from 61.45%(training on shortest-path trajectories) to 65.35%(training on self-demonstration trajectories) and 65.45%(training on better self-demonstration trajectories). Although SR converges in the third round, there's an additional gain of SPL(43.31% to 44.62%). **Without running over 2 rounds, we cannot confirm convergence or justify stopping early.**
> - **Scale up with additonal 60 HM3D envs:** We conduct two rounds to demonstrate the effective scalability of SID with larger training environments. As in Table 4, SR on the validation unseen split of image goal navigation improves from 65.45%(60 training environments) to 69.9% (120 training environments, round1), then to 70.75% (120 training environments, round2). **A single iteration is insufficient to verify convergence or demonstrate that SID retains its self-improving nature as data scale increases.**
> - **Scale up with total 800 HM3D envs: We run only one self-improvement round due to compute cost**, yielding +10% SR and SPL as shown in Table 3.

---

> ### Author Response · Authors · 2025-11-22
> **Response to Reviewer DHqx - Part 3**
>
> >**W2: Cost and accessibility.**
>
>  We acknowledge the compute cost but emphasize its necessity and efficiency:
>
> - SID provides a **data-centric** contribution—the first large-scale demonstration on exploration strategies for goal-oriented VLN, filling a critical gap in this field. Compared to previous manually annotated method(e.g. [Habitat-Web](https://ram81.github.io/projects/habitat-web.html)), our resource use is reasonable for the scalability and reusability.
> - Our captioning cost is modest: **~2M image description instructions** → **46M (instruction, trajectory)** pairs via rollout. By contrast, [SRDF](https://iclr.cc/virtual/2025/poster/29824) generates 20M instructions using Mantis (SigLIP + LLaMA-3), implying ~10× higher captioning compute for only half the training corpus.
> - Pretraining compute: [Uni-NaVid](https://arxiv.org/pdf/2412.06224) utilizes 1,400 NVIDIA H800 GPU-hours and [StreamVLN](https://arxiv.org/pdf/2507.05240) uses 1,500 NVIDIA A100 GPU-hours. Our computational cost is reasonable since our DUET model is pretrained on 46M large-scale instruction-trajectory pairs.
> - The one-time data generation of SID can support diverse navigation agents without further processing or environment interaction.
>
> [1] Ramrakhya R, Undersander E, Batra D, et al. Habitat-web: Learning embodied object-search strategies from human demonstrations at scale[C]//Proceedings of the IEEE/CVF conference on computer vision and pattern recognition. 2022: 5173-5183.
>
> [2] Wang Z, Li J, Hong Y, et al. Bootstrapping language-guided navigation learning with self-refining data flywheel[J]. arXiv preprint arXiv:2412.08467, 2024.
>
> [3] Zhang J, Wang K, Wang S, et al. Uni-navid: A video-based vision-language-action model for unifying embodied navigation tasks[J]. arXiv preprint arXiv:2412.06224, 2024.
>
> [4] Wei M, Wan C, Yu X, et al. Streamvln: Streaming vision-and-language navigation via slowfast context modeling[J]. arXiv preprint arXiv:2507.05240, 2025.

---

> ### Author Response · Authors · 2025-11-22
> **Response to Reviewer DHqx - Part 4**
>
> >**W4: Analysis of the ablations.**
>
> In fact, our paper already provides detailed analysis regarding the reviewer's concern of ablations.
>
> - **Isolation of “teacher-forcing” vs “student-forcing”:**
>   - **The definition of teacher-forcing and student-forcing is stated in line 255 of the paper:** "We interleave teacher forcing with demonstrations (where the agent strictly follows the given trajectories) and student forcing with shortest-path oracle actions (where the agent executes its own predicted actions but is supervised by the oracle)." **We compare different supervision strategies in teacher forcing in the Table 6:** following the shortest paths versus following exploration paths. **These distinct strategies correspond to the isolation:** if teacher forcing employs only shortest-path trajectories, it effectively reduces to applying shortest-path–based student forcing alone—thereby eliminating the contribution of exploratory behavior in the demonstration phase.
>   - As demonstrated in Table 6, **teacher forcing with self-demonstrations benefits finetuning** across different data scales and goal modals.  Even with the same shortest-path demonstrations for finetuning, the image-goal navigation agent **pretrained with self-demonstrations (Table 6, line 1)** still performs significantly better than the one pretrained with shortest-path demonstrations (Table 3, line 1). This further underscores that such **self-demonstrations benefit both pretraining and finetuning.**
> - **Trajectory-quality stratification:** We conduct our analysis in the line 413 of our paper. We sample random paths for goal-oriented navigation data, which is similar to the distribution of the agent’s exploration ones.  As claimed in line 416, Table 7 shows that the agent trained on the randomly sampled trajectories can only reach the equivalent performance with the shortest path but far behind the exploration paths of the agent itself, which strongly claims that **the agent learns effectively from its own exploration rather than arbitrary trajectories.**
> - We also demonstrate that **the primary benefit arises from the SID paradigm rather than simply introducing more data in new environments**, as shown in Table 8.
> - **The causal story behind SID’s improvements:**  In line 395, **we give a detailed analysis of the room exploration and error correction capbilities of the navigation agent:** the agent trained on self-exploration data explores a greater number and variety of rooms than the shortest paths. Besides, the number of rooms explored of the same type as the target has increased, **verifying that the agent actively explores similar rooms and corrects its wrong steps to reach the correct destination.** We also provide visualization of the navigation agent’s exploration trajectories and the predicted actions at each step at the end of the Appendix.

---

> ### Author Response · Authors · 2025-11-22
> **Response to Reviewer DHqx - Part 5**
>
> >**Q3&W3: Transfer beyond indoor graphs:**
>
> We have conducted additional experiments to demonstrate that **the exploration priors learned via SID can effectively translate to continuous control, a critical step that closely aligns with real-world deployment.**
>
> - Given that [VLN-CE](https://jacobkrantz.github.io/vlnce/) is the standard publicly available benchmark for continuous vision-language navigation, we utilized this environment for our evaluation. Specifically, we compare the navigation agent pretrained on our SID-VLN data against the one pretrained on ScaleVLN-RVR, which is the goal-oriented navigation version of the widely-used [ScaleVLN](https://scalevln.github.io/), making it a strong baseline for a controlled comparison due to the similarity in task focus with SID-VLN data. To bridge the gap between discrete and continuous actions, both models are finetuned utilizing the waypoint predictor proposed by [Discrete-Continuous-VLN](https://openaccess.thecvf.com/content/CVPR2022/papers/Hong_Bridging_the_Gap_Between_Learning_in_Discrete_and_Continuous_Environments_CVPR_2022_paper.pdf). The detailed performance on the Validation Unseen split of VLN-CE is presented below (NE denotes Navigation Error):
>
>   | Pretraining Data   | NE($\downarrow$) | SR($\uparrow$) | SPL($\uparrow$) |
>   | ------------------ | ---------------- | -------------- | --------------- |
>   | ScaleVLN-RVR       | 5.44             | 44             | 39              |
>   | **SID-VLN (Ours)** | **5.16**         | **47**         | **41**          |
>
>   **The agent pretrained on SID-VLN consistently outperforms the baseline pretrained on ScaleVLN-RVR, empirically showcasing better generalization to cotinuous navigation**. Although full deployment on real robots is impractical within the tight timeline of the rebuttal period due to the substantial engineering effort, we will remain committed to pursuing real-robot deployment and include these results in the final version of our paper following your advice.
>
> [1] Wang Z, Li J, Hong Y, et al. Scaling data generation in vision-and-language navigation[C]//Proceedings of the IEEE/CVF international conference on computer vision. 2023: 12009-12020.
>
> [2] Hong Y, Wang Z, Wu Q, et al. Bridging the gap between learning in discrete and continuous environments for vision-and-language navigation[C]//Proceedings of the IEEE/CVF conference on computer vision and pattern recognition. 2022: 15439-15449.
>
> [3] Krantz J, Wijmans E, Majumdar A, et al. Beyond the nav-graph: Vision-and-language navigation in continuous environments[C]//European Conference on Computer Vision. Cham: Springer International Publishing, 2020: 104-120.

---

### Author Response · Authors · 2025-11-24
**General Response**

We extend our sincere gratitude to all the reviewers, AC, PC, and SPCs  for dedicating their time and effort to evaluating our paper. We also thank the reviewers for their positive and insightful comments, which can help us improve our work.

**Summary of Reviewer Positive Feedback**

- **Scalability and Coverage:** All the reviewers (DHqx, geo2, xvKU, and DXJg) think that SID generates 46M language-goal exploration trajectories across ~860 environments, which is the first large-scale demonstrations on exploration strategies and tackles the data scarcity in goal-oriented VLN.
- **State-of-the-art Performance:** All the reviewers agree that SID achieves state-of-the-art results on REVERIE and SOON, outperforming recent large-scale baselines under a shared HM3D-800 setting.
- **Methodological Novelty:** Reviewer DHqx and DXJg recognize that SID introduces a novel self-improving exploration-driven scalable paradigm that evolves exploration trajectories rather than instructions, which enables the navigation agent to learn exploration priors from its own demonstrations and showcases transferable learning across datasets.
- **Reproducibility and Clarity:** Reviewer DHqx and DXJg think that our paper offers clear, reproducible implementation details—including model choices and GPU-hour breakdowns—and is accessibly written for a broad audience.

**Summary of Key Revisions and Discussion**

- For Reviewer geo2 and xvKU, we provide a detailed analysis of our novelty and unique contribution, including path evolution paradigm, large-scale demonstrations generation, profound ablation studies and unified goal-oriented navigation enhancement.
- For Reviewer DHqx and DXJg, we present results of transferring SID towards continuous-environment vision-language-navigation, which more closely aligns with real-world deployment.
- For Reviewer DHqx, we clarify the analysis of caption style impact stated in our paper.
- For Reviewer DHqx, we justify the value of our SID round and computation cost.
- For Reviewer DHqx, we showcase the detailed ablation studies in the paper, especially the isolation of “teacher-forcing” vs “student-forcing”  and trajectory-quality stratification.
- For Reviewer geo2, we analyze why MLLMs are unsuitable as base navigation models in our pipeline due to their poor performance in goal-oriented navigation and prohibitively high inference costs.
- For Reviewer geo2, we discuss immense domain disparity between embodied navigation and GUI/tool-use agent, and justify the the foundational role of goal-oriented navigation.
- For Reviewer xvKU, we explain why online reinforcement learning is impractical in VLN, citing issues such as historical evidence, empirical validation and theoretical analysis.
- For Reviewer DXJg, we clarify the general experimental settings and the reviewer's confusion underlying our work.
- For Reviewer DXJg, we explain why direct comparisons with methodologically disparate approaches are not feasible and present the policy transfer comparison in our paper to bridge this gap.

We appreciate all reviewers’ feedback and remain committed to further improving the work. The following sections address each reviewer’s comments in detail. We kindly invite you to review our rebuttal and let us know if our responses have adequately addressed your concerns. If you have any additional questions or feedback, we would be glad to further discuss and clarify them.

---

### Author Response · Authors · 2025-11-25
**Kind Request to Review Our Rebuttal**

Dear Reviewers,

Thank you sincerely for the time and effort in reviewing. In our rebuttal, we have provided detailed responses to the weaknesses and questions you raised. We also present experiments of transferring SID towards continuous vision-language-navigation. As the discussion deadline approaches in a week, we would like to kindly ask whether you have any additional questions or if there are any further experiments you would like us to consider. We are truly happy to provide further clarification for any points of concern. Your feedback is truly valuable for improving our work, and we would be very grateful if we could receive your reconsideration of our submission.

Best regards,

Authors

---

### Author Response · Authors · 2025-12-03
**Thank AC for taking over our submission and Summary of Review-Rebuttal phase (1/2)**

Thank you for taking on this paper and for being invited to participate in our discussion with the reviewers. We understand that the special circumstances this year have placed an additional burden on ACs, and we genuinely appreciate your time and effort. **To help you efficiently assess our work, we provide a concise summary of our contribution, the reviewers' evaluation, and the key improvements made during rebuttal.**


>**Our main contribution**

- We present **Self-Improving Demonstrations (SID)**, the first approach that enables a navigation agent to learn robust exploration strategies with demonstrations from its own successful trajectories, averting the dependence on costly human annotations for exploration data.
- We provide large-scale exploratory language-goal trajectories along with a capable navigation agent that can be used to generate additional exploration paths, which is the first to offer transferable large scale demonstrations on exploration strategies, filling a critical gap in goal-oriented navigation.
- We demonstrate SID-VLN’s scalability across diverse environments and transferability to goal-oriented language-guided navigation tasks through vision-language-aligned pretraining, achieving state-of-the-art results on challenging goal-oriented VLN tasks including SOON and REVERIE.
- We provide in-depth component ablation studies detailing the contribution of each module in our method, demonstrating insights for general self-evolving agent learning.


>**Strengths recognized by the reviewers**

- **Methodological Novelty:** SID-VLN introduces a novel self-improving exploration-driven scalable paradigm that evolves exploration trajectories rather than instructions, which enables the navigation agent to learn exploration priors from its own demonstrations and showcases transferable learning across datasets. (DHqx, DXJg)
- **Scalability and Coverage:** SID-VLN generates 46M language-goal exploration trajectories across 860 environments, which is the first large-scale demonstrations on exploration strategies and tackles the data scarcity in goal-oriented VLN. (DHqx, geo2, xvKU, DXJg)
- **State-of-the-art Performance:** SID-VLN achieves state-of-the-art results on REVERIE and SOON, outperforming other large-scale baselines under the shared setting. (DHqx, geo2, xvKU, DXJg)
- **Reproducibility and Clarity:** The paper offers clear, reproducible implementation details and is accessibly written for a broad audience. (DHqx, DXJg)

---

> ### Author Response · Authors · 2025-12-03
> **Thank AC for taking over our submission and Summary of Review-Rebuttal phase (2/2)**
>
> >**Key improvements and clarification made during rebuttal**
>
> - **Clarification of Novelty and Methodology:** Presented the unique contribution of the "path evolution" paradigm and large-scale demonstration generation. (geo2, xvKU) Provided a comprehensive analysis (historical evidence, empirical validation and theoretical analysis) justifying why online reinforcement learning with a simple sparse success reward is not suitable for VLN. (xvKU)
> - **Baseline Selection and Domain Scope:** Explained the unsuitability of current MLLMs as base agents due to high inference costs and poor goal-oriented navigation performance. Discussed the significant domain disparity and infeasibility of integration between embodied navigation and GUI tool-use agents. Justified the the foundational role of goal-oriented navigation. (geo2)
> - **Transfer to Continuous Environments:** Presented new experimental results transferring SID to continuous-control VLN (which closely aligns with real-world deployment), demonstrating significantly stronger generalization than the baseline. (DHqx, DXJg)
> - **Review of Mechanism Analysis in the paper:** Expanded on detailed ablation studies, specifically isolating the effects of "teacher-forcing" vs. "student-forcing," trajectory-quality stratification, and the impact of different caption styles. (DHqx)
> - **Computational Efficiency and Experimental Settings:** Justified the value of iterative SID rounds relative to computational costs. (DHqx) Clarified general experimental settings to resolve reviewer confusion. Explained the infeasibility of direct comparisons with methodologically disparate approaches and presented the policy transfer comparison in our paper to bridge this gap. (DXJg)
>
>
> >**Reviewer Endorsement and Request to Area Chair**
>
> We sincerely appreciate the constructive feedback from the reviewers. We note that **Reviewers DHqx and DXJg initially leaned towards acceptance (Rating: 6).** Furthermore, **Reviewer xvKU explicitly stated a willingness to raise the score** provided that our unique contributions were clarified—a point we have comprehensively addressed in our rebuttal.
>
> We understand that the recent technical issues on OpenReview may have impacted the reviewers' ability to fully engage in the discussion phase. As a result, we were only able to exchange further correspondence with Reviewer DXJg. **Given that the reviewers indicated a confidence score of 3, we would deeply appreciate it if an Area Chair with expertise in Vision-Language Navigation domain could facilitate the final decision.** We trust that a domain expert is best positioned to evaluate the validity of our rebuttals regarding goal-oriented VLN and our methodological contributions.
>
> We hope this summary facilitates your evaluation and remain available for any clarifications.
>
> Best regards,
>
> The Authors

---

### Meta-Review · Area_Chair_nY1v · 2026-01-10

**Summary:**

This paper proposes an iterative learning + self-improvement loop where successful rollouts from the trained model are used to improve the model, with transfer to language-conditioned trajectories via VLM captioning. The ratings were mixed (2,4,6,6), with a number of concerns raised with a number of important ones that speak to the situation of the method in terms of the literature and compute/improvement tradeoff. For example, Reviewer DHqx mentions lack of analysis in terms of marginal gains of additional SID rounds. Reviewer geo2 points out that this self-generated trajectory loop is standard across a number of uni-modal and multi-modal tasks, yet the results are only on VLN domains. Reviewer xvKU mentions that the proposed method seems overly complex, compared to similar exploration/exploitation learning that is intrinsic to reinforcement learning methods. Reviewer DXJg had one of the more positive scores but had an in-depth discussion about the fact that VLN is the only focus, with exploration being a part of the task but can also be potentially informed by contextual cues instead in the real world, as well as comparisons to additional baselines and claims regarding other methods' use of human demonstrations.

**Reviewer Concerns:**

Some concerns were addressed well by the rebuttal. For example, results were shown for continuous-environment VLN, analysis in terms of compute/improvement were provided, and MLLMs were tried as baselines showing that they cannot provide good base performance that can drive self-improvement. Other concerns, though, mentioned above (some of which were mentioned by multiple reviewers) were not well-addressed. This includes better situation of the method with respect to other exploration algorithms including reinforcement learning (which has a large body of work across many domains) and concerns about why this method is well-suited to VLN and not demonstrated across other domains.

**Reviewer Scores:**

Reviewer xvKU, who had a score of 2, mentioned the possibility of raising their score on "clearer evidence of its unique contributions", and especially compared to reinforcement learning. While the authors provide a rebuttal based on historical evidence, empirical validation, and "theoretical" analysis, they are not overall convincing. The theoretical analysis (which isn't theoretical at all) just says that sparse RL reward make it so that the agent "cannot identify which actions led to success". However, this is indeed the entire point of reinforcement learning algorithms over expectation, and while it is a difficult problem "cannot" is not accurate. The historical evidence hinges on GPU hours required for RL training, yet the SID method itself is also much more computationally costly. Finally, the empirical validation is preliminary and ignores a range of SFT/IL initialization + RL finetuning methodologies across not just the RL literature but VLN/Vision-Language-Action (VLA) literature. Further, sparse rewards are not the only possibility (as mentioned by the authors in terms of reward shaping). Overall, while the proposed method does yield good performance, it is necessary to thoroughly situate the method with respect to strong state of art RL baselines as it is not clear that the compute (and pipeline complexity) versus improvement tradeoff is any better for the proposed method.

Reviewer geo2 questioned the novelty of the work, lack of generalization to other application domains beyond VLN, and lack of use of state of art LLMs. While the last concern was addressed with some justification, the overall contribution statement is not strongly likely to have changed the reviewer's mind. As an example, the statement justifying the uniqueness of VLN for SID methods ("Goal oriented navigation requires active exploration and spatial error correction in 3D environments, which cannot be solved by simple pattern matching") has many issues. First, exploration is definitely present in a large number of other domains, ranging from code/math (which can be combined with visual data as well) to Computer Use agents. While not all "exploration" in those domains are spatial in nature, exploring in concept/search space has similar characteristics. Further, the method proposed is not uniquely positioned for 3D as it doesn't natively use it in the model (as other reviewers have mentioned, there are 3D/geometric exploration techniques as well).

Given the paper, reviews, rebuttal, and (limited) discussion, overall it is clear that there are still a number of important outstanding concerns that were not well-addressed in the rebuttal and require substantial experimentation and validation, namely comprehensive comparison to a range of state of art RL methods and a better situation of the method to the specific domain that was demonstrated (VLN) or else validation in other domains.

---

### Decision · Program_Chairs · 2026-01-26

Reject